# Pseudo-Riemannian Graph Transformer

**Viet Quan Le**[1]  **Viet Cuong Ta**[2]*

[1]Department of Computer Science   [2]Human-Machine Interaction Laboratory
VNU University of Engineering and Technology, Hanoi, Vietnam
{lvquan, cuongtv}@vnu.edu.vn

## Abstract

Many real-world graphs exhibit diverse and complex topological structures that are not well captured by geometric manifolds with uniform global curvature, such as hyperbolic or spherical spaces. Recently, there has been growing interest in embedding graphs into pseudo-Riemannian manifolds, which generalize both hyperbolic and spherical geometries. However, existing approaches face three significant limitations, including the ineffective pseudo-Riemannain framework, the shallow architectures, and the absence of clear guideline for selecting suitable pseudo-Riemannian manifolds. To address these issues, we introduce a novel diffeomorphic framework for graph embedding that aligns with the nature of pseudo-Riemannian manifolds. Subsequently, we propose the pseudo-Riemannian Graph Transformer for learning representations of complex graph structures. Our diffeomorphic framework in pseudo-Riemannian geometry enables the principled definitions of core transformer components, including linear attention, residual connection, and layer normalization. Finally, we develop a lightweight space searching algorithm to automatically identify the most suitable pseudo-Riemannian manifold for an input graph. Extensive experiments on diverse real-world graphs demonstrate that our model consistently outperforms other baselines in both node classification and link prediction tasks.

## 1 Introduction

Many real-world graphs have diverse topological structures (Boguñá et al., 2021), which are better captured using varying geometrical curvatures. Geometric manifolds with uniform global structure, such as hyperbolic or spherical manifolds, do not provide enough flexibility to model these complex graphs (Gu et al., 2018). Instead of using a single manifold with a constant curvature, researchers employ the Cartesian product of constant curvature manifolds to embed graphs of mixed topologies (Gu et al., 2018; Bachmann et al., 2020). However, explicit distances in product manifolds may emphasize either the spherical or hyperbolic component while overlooking the other, which limits their ability to embed heterogeneous graphs (Law, 2021). Moreover, because Riemannian manifolds like spherical manifolds use a positive definite metric, they cannot accurately capture the negative eigenvalues present in input similarity data (Laub and Müller, 2004).

Recently, pseudo-Riemannian manifolds (also called semi-Riemannian manifolds), which generalize Riemannian manifolds, have gained increasing attention for embedding mixed topological graphs (Law and Stam, 2020; Law, 2021; Xiong et al., 2022). Pseudo-Riemannian manifolds equipped with indefinite metrics not only extend hyperbolic and spherical geometries but also include hyperbolic and spherical submanifolds, allowing them to capture relationships unique to these geometries. Law and Stam (2020) proposed an efficient method to learn nonparametric embeddings with a novel optimization tool on pseudo-Riemannian manifolds. Later, Law (2021) employed pseudo-Riemannian geometry to graph neural networks by using the quotient manifold, which imposes

---

*Corresponding author.

39th Conference on Neural Information Processing Systems (NeurIPS 2025).

additional constraints on the optimized function. Xiong et al. (2022) introduced the diffeomorphic framework which decomposes pseudo-Riemannian manifolds into the product of spherical and Euclidean spaces. This framework helps avoid *geodesic disconnection* and simplifies the extension of GCNs to pseudo-Riemannian geometry. Compared to the work of Law (2021), the diffeomorphic framework offers greater flexibility, as it can be applied to any optimized function within pseudo-Riemannian geometry. Nevertheless, existing methods exhibit three major drawbacks that limit their learning capacity. First, since pseudo-Riemannian manifolds inherently comprise both spherical and hyperbolic submanifolds, this transformation onto the product of spherical and Euclidean spaces (Xiong et al., 2022) results in a loss of hierarchical information essential to hyperbolic geometry. Second, there is no clear guideline for selecting the number of time and space dimensions for pseudo-Riemannian approaches. A mismatch between the graph structure and the topology of the embedding manifold can introduce distortions in the graph representation (Ganea et al., 2018; Chami et al., 2019; Bachmann et al., 2020). Third, the context size in GNN architectures is often limited (Li et al., 2018; Oono and Suzuki, 2020). As the depth of GNNs increases, it leads to a phenomenon known as *oversmoothing* or *oversquashing* (Chen et al., 2020) where node representations become indistinguishable across the graph. To mitigate this issue, recent studies have introduced Transformer-based graph encoders in Riemannian geometry (Wu et al., 2021; Cho et al., 2023; Yang et al., 2024). However, their adaptation to more general Riemannian-based geometries has not yet been studied because developing key Transformer operations on such manifolds is not trivial. Unlike Riemannian manifolds, pseudo-Riemannian manifolds may contain pairs of points that cannot be connected by a continuous geodesic (Law, 2021). As a result, gradient vectors are not parallel transported along geodesics, and the logarithmic map between two given points is undefined.

In this work, we introduce a diffeomorphism that decomposes pseudo-Riemannian manifolds into the Cartesian product of spherical and hyperbolic spaces. This transformation, which is built upon a submanifold with a smooth extrinsic mapping function, addresses the issue of geodesic disconnection and effectively preserves the cyclical and hierarchical structures of graphs. Subsequently, we propose an efficient pseudo-Riemannian Graph Transformer ($\mathcal{Q}$-GT) for learning representations of complex graphs. Our diffeomorphic framework enables the definition of fundamental operations, including linear attention, residual connection, and layer normalization, while allowing the use of standard gradient descent algorithms for effective optimization. Additionally, we develop a lightweight space searching algorithm to determine the optimal pseudo-Riemannian manifold for representing mixed-topology graphs. To achieve this, we construct a bijection that maps each pseudo-Riemannian manifold to an *ideal* Gaussian sectional curvature (GSC) distribution, and apply the Kullback–Leibler (KL) divergence to select a suitable manifold. Finally, comprehensive evaluations on node classification and link prediction tasks show that our model surpasses both Riemannian and pseudo-Riemannian methods.

**Contributions.** Our contributions in this paper include: 1) We introduce a novel diffeomorphic framework that transforms pseudo-Riemannian manifolds into the product of spherical and hyperbolic space to preserve the cyclical and hierarchical representations of graphs. 2) We propose an efficient pseudo-Riemannian Graph Transformer ($\mathcal{Q}$-GT) for capturing mixed-topological dependencies of input graphs. To the best of our knowledge, this is the first study on a Transformer-based graph model built within the pseudo-Riemannian geometry. 3) We develop an efficient and effective space searching algorithm for determining the time and space hyperparameters of pseudo-Riemannian manifolds based on sectional curvature distributions.

## 2 Preliminaries

A generalization of Riemannian manifold is pseudo-Riemannian manifold (O'neill, 1983) which allows the metric tensor $g$ to be indefinite, meaning that for any nonzero tangent vector $\xi \in \mathcal{T}_x\mathcal{M}$, the value of $g_x(\xi, \xi)$ can take on both positive and negative signs. The ambient pseudo-Euclidean space $\mathbb{R}^{p,q+1}$ is a $d$-dimensional vector space, where $d = p + q + 1 \in \mathbb{N}$, equipped with the following scalar product $\langle x, y \rangle_q = -\Sigma_{i=0}^{q} x_i y_i + \Sigma_{j=q+1}^{q+p} x_j y_j \; \forall x, y \in \mathbb{R}^{p,q+1}$. A pseudo-hyperboloid $\mathcal{Q}_{\beta}^{p,q}$ is defined as the submanifold in the ambient pseudo-Euclidean space (Law and Stam, 2020):

$$\mathcal{Q}_{\beta}^{p,q} = \{x = (x_0, x_1, ..., x_{p+q}) \in \mathbb{R}^{p,q+1} : \|x\|_q^2 = \beta\}, \tag{1}$$

where $\beta$ is a nonzero real number indicating space curvature, and $\|x\|_q^2 = \langle x, x \rangle_q$ is the vector norm in the pseudo-Euclidean space. $\mathcal{Q}_{\beta}^{p,q}$ is called *pseudo-sphere* when $\beta > 0$ and *pseudo-hyperboloid*

when $\beta < 0$. Since $\mathcal{Q}_\beta^{p,q}$ and $\mathcal{Q}_{-\beta}^{q+1,p-1}$ via an anti-isometry, we consider the pseudo-hyperboloid in this work. Following the terminology of space-time vectors (Sun et al., 2015), for each point in $\mathcal{Q}_\beta^{p,q}$, the first $q+1$ dimensions are time dimensions and the last $p$ dimensions are space dimensions. Hyperbolic and spherical manifolds can be considered as special cases of pseudo-hyperboloid $\mathcal{Q}_\beta^{p,q}$. The $d$-dimensional spherical manifold, defined as $\mathbb{S}_{|\beta|}^q = \{x \in \mathbb{R}^{q+1} : \langle x, x \rangle = |\beta|\}$ where $\langle ., . \rangle$ is the standard inner product in Euclidean space, is a pseudo-hyperboloid $\mathcal{Q}_\beta^{0,q}$. Meanwhile, the $p$-dimensional Lorentz model $\mathbb{L}_\beta^p$ (Chami et al., 2019), which is used to represent hyperbolic manifolds, is an upper sheet of the two-sheet pseudo-hyperboloid $\mathcal{Q}_\beta^{p,0}$. The Lorentz model is defined as $\mathbb{L}_\beta^p = \{x \in \mathbb{R}^{p,1} : \langle x, x \rangle_\mathbb{L} = \beta, x_0 > 0\}$, where $\langle ., . \rangle_\mathbb{L} = \langle ., . \rangle_{q=0}$ denotes the Lorentzian inner product. To build a graph Transformer in pseudo-Riemannian geometry, we rely on further geodesic tools in the pseudo-hyperboloid model, including the exponential and logarithmic maps along with the parallel transport.

**Exponential and logarithmic maps.** Given a reference point $x \in \mathcal{Q}_\beta^{p,q}$, the maps between the manifold $\mathcal{Q}_\beta^{p,q}$ and the tangent space $\mathcal{T}_x \mathcal{Q}_\beta^{p,q}$ include the exponential map $\exp_x^\beta : \mathcal{T}_x \mathcal{Q}_\beta^{p,q} \to \mathcal{Q}_\beta^{p,q}$ and the logarithmic map $\log_x^\beta : \mathcal{Q}_\beta^{p,q} \to \mathcal{T}_x \mathcal{Q}_\beta^{p,q}$. Law and Stam (2020) proposed the closed-form expressions of two tangent maps, which we provide in the Appendix D for reference. They noticed that a geodesic does not exist between two points $x$ and $y$ if the scalar product satisfies $\langle x, y \rangle_q \geq |\beta|$. Consequently, the logarithmic map is undefined at $y$, and the points $x$ and $y$ are considered *geodesic-disconnected*.

**Parallel transport.** Given two points $x$ and $y$ that are connected by a geodesic $\gamma$ on the manifold $\mathcal{Q}_\beta^{p,q}$, parallel transport $P_{x \to y}^\gamma : \mathcal{T}_x \mathcal{Q}_\beta^{p,q} \to \mathcal{T}_y \mathcal{Q}_\beta^{p,q}$ is an isomorphism between two tangent spaces that moves one tangent vector $\xi \in \mathcal{T}_x \mathcal{Q}_\beta^{p,q}$ to another tangent space $\mathcal{T}_y \mathcal{Q}_\beta^{p,q}$ along the geodesic $\gamma$ (Law, 2021). The formula of the parallel transport is given in the Appendix D.

## 3 Diffeomorphic framework

Based on the pseudo-hyperboloid formulation, we propose a diffeomorphic framework that decomposes a pseudo-Riemannian manifold into the product of spherical and hyperbolic spaces. This framework effectively preserves the graphs' cyclical and hierarchical structures while addressing the geodesic disconnections issue. Our framework is centralized by an extrinsic mapping function $\Phi$ which transports points from a manifold $\mathcal{Q}_\beta^{p,q}$ to a sub-manifold $\hat{\mathcal{Q}}_\beta^{p,q+1}$, followed by a diffeomorphism $\Psi$ that transforms $\hat{\mathcal{Q}}_\beta^{p,q+1}$ to the product of a sphere $\mathbb{S}_{|\beta|}^q$ and a Lorentz model $\mathbb{L}_\beta^p$.

### 3.1 Extrinsic mapping function

To construct an extrinsic mapping function, we first define a *zero-first submanifold* $\hat{\mathcal{Q}}_\beta^{p,q}$ of a pseudo-hyperboloid $\mathcal{Q}_\beta^{p,q}$ as follows:

**Definition 1** (*Zero-first submanifold*). *A zero-first submanifold* $\hat{\mathcal{Q}}_\beta^{p,q}$ *of a pseudo-hyperboloid* $\mathcal{Q}_\beta^{p,q}$ *is defined as*

$$\hat{\mathcal{Q}}_\beta^{p,q} = \{x \in \mathbb{R}^{p,q+1} : \|x\|_q^2 = \beta, x_0 = 0\}. \tag{2}$$

Then the extrinsic mapping function $\Phi : \mathcal{Q}_\beta^{p,q} \to \hat{\mathcal{Q}}_\beta^{p,q+1}$ and its inverse $\Phi^{-1} : \hat{\mathcal{Q}}_\beta^{p,q+1} \to \mathcal{Q}_\beta^{p,q}$ are formulated as:

$$\Phi(x) = (0, x) = y \,, \Phi^{-1}(y) = x, \tag{3}$$

where $x$ is a point on pseudo-hyperboloid manifold $\mathcal{Q}_\beta^{p,q}$, and $y$ is a corresponding point on the zero-first submanifold $\hat{\mathcal{Q}}_\beta^{p,q+1}$. Several key properties of the extrinsic function $\Phi$ are presented in the following proposition.

**Proposition 1.** *The extrinsic mapping function $\Phi$ and its inverse $\Phi^{-1}$ are smooth, bijective, and isometric.*

The smoothness of the extrinsic mapping allows it to be integrated into the Graph Transformer module in pseudo-Riemannian spaces. Furthermore, its bijective and isometric properties help preserve the

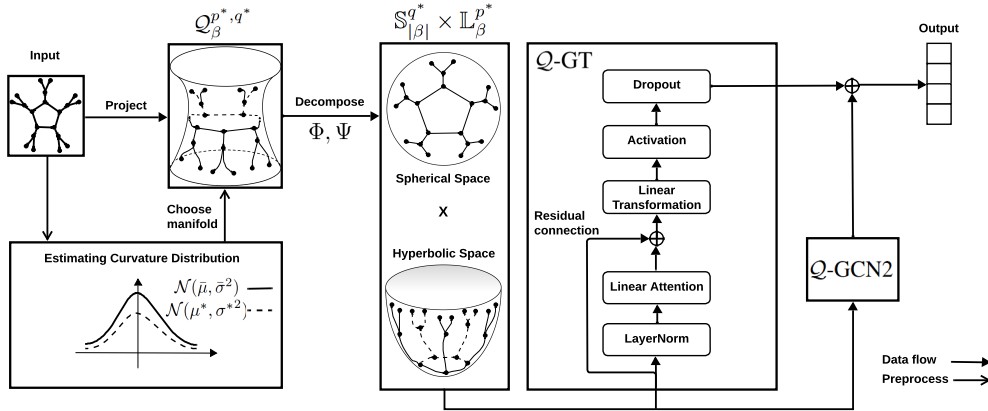

Figure 1: The overall architecture of our proposed $\mathcal{Q}$-GT.

integrity of original node features during vector addition on the pseudo-hyperboloid, addressing limitations in previous works (Law, 2021; Xiong et al., 2022).

## 3.2 Diffeomorphic function

The following theorem provides a diffeomorphism that transforms the submanifold $\hat{\mathcal{Q}}_\beta^{p,q+1}$ into the product of a spherical manifold $\mathbb{S}_{|\beta|}^q$ and a hyperbolic manifold $\mathbb{L}_\beta^p$.

**Theorem 1.** *For any point $x \in \hat{\mathcal{Q}}_\beta^{p,q+1}$, there exists a diffeomorphism $\Psi : x \in \hat{\mathcal{Q}}_\beta^{p,q+1} \to \mathbb{S}_{|\beta|}^q \times \mathbb{L}_\beta^p$ that maps $x$ onto the product manifolds of a sphere and a hyperboloid. The mapping and its inverse are given by:*

$$\Psi(x) = \begin{pmatrix} \sqrt{|\beta|}\frac{u}{\|u\|} \\ \|u\| \\ v \end{pmatrix} , \ \Psi^{-1}(z) = \begin{pmatrix} 0 \\ \frac{v_0'}{\sqrt{|\beta|}}u' \\ v_{[1:]}' \end{pmatrix}, \tag{4}$$

*where $x = \begin{pmatrix} 0 \\ u \\ v \end{pmatrix} \in \hat{\mathcal{Q}}_\beta^{p,q+1}$ with $u \in \mathbb{R}^{q+1} \setminus \{\mathbf{0}\}$, $v \in \mathbb{R}^p$, and $z = \begin{pmatrix} u' \\ v' \end{pmatrix}$ with $u' \in \mathbb{S}_{|\beta|}^q$, $v' \in \mathbb{L}_\beta^p$.*

The notation $\|u\| = \sqrt{\langle u, u \rangle}$ indicates the Euclidean norm. The proof of **Theorem 1** is given in the Appendix E. Compared to the diffeomorphism proposed by Xiong et al. (2022), our framework better preserves the hierarchical structure of graphs. The approach in Xiong et al. (2022) transforms points on the pseudo-hyperboloid into a more Euclidean product space as the space dimension $p$ increases, potentially diminishing the hyperbolic properties essential for capturing hierarchical relationships. Meanwhile, our diffeomorphism maps points on the pseudo-hyperboloid to a product of spherical and hyperbolic spaces that aligns naturally with two geometric properties of pseudo-Rienammian hyperboloid: (**i**) containing spherical and hyperbolic topologies, (**ii**) becoming a spherical (or hyperbolic) manifold when $p = 0$ (or $q = 0$). Hence, the product space $\mathbb{S}_{|\beta|}^q \times \mathbb{L}_\beta^p$ can be used to interpret the evolution of pseudo-hyperboloid $\mathcal{Q}_\beta^{p,q}$, leading to our efficient space searching algorithm for determining the optimal pseudo-hyperboloid in Section 5. The proposed diffeomorphic framework can be viewed as a generalization of the hyperbolic framework that underlies state-of-the-art non-Euclidean methods such as HGCN (Chami et al., 2019) and Hypformer (Yang et al., 2024).

## 4 Pseudo-Riemannian Graph Transformer

To address the limitations of pseudo-Riemannian GNNs, we propose an efficient pseudo-Riemannian Graph Transformer ($\mathcal{Q}$-GT), as illustrated in Fig. 1. By leveraging our diffeomorphic framework in Section 3, we extend fundamental operations, including linear attention, and layer normalization to the pseudo-hyperboloid $\mathcal{Q}_\beta^{p,q}$. Additionally, we design a novel residual connection that operates directly on the pseudo-hyperboloid, ensuring stable training and enhancing the performance of $\mathcal{Q}$-GT.

## 4.1 Pseudo-Riemannian linear attention

Given the node features $X^{\mathcal{Q}_\beta^{p,q}} = \{x_i^{\mathcal{Q}_\beta^{p,q}}\}_{i=1}^{i=N}$ of input graph $G$, where each $x_i^{\mathcal{Q}_\beta^{p,q}} \in \mathcal{Q}_\beta^{p,q}$, we first apply the extrinsic mapping function $\Phi$ and the diffeomorphism $\Psi$, as defined in Eq. (3) and Eq. (4), to project the features onto the product space, i.e. $X^{\mathbb{S}_{|\beta|}^q \times \mathbb{L}_\beta^p} = \Psi(\Phi(X^{\mathcal{Q}_\beta^{p,q}}))$.

We decompose each node feature $x_i^{\mathbb{S}_{|\beta|}^q \times \mathbb{L}_\beta^p} = (u_i, v_i)^T$ where $u_i \in \mathbb{S}_{|\beta|}^q$ and $v_i \in \mathbb{L}_\beta^p$. Since the spherical and hyperbolic manifolds are independent within the product space $\mathbb{S}_{|\beta|}^q \times \mathbb{L}_\beta^p$, we can apply linear attention to each manifold separately and then concatenate the results. Yang et al. (2024) proposed a linear attention mechanism in hyperbolic space $\mathbb{L}_\beta^p$, allowing for efficient processing of large-scale data. Specifically, the hyperbolic linear attention can be described as:

$$v_i^W = \left( \sqrt{\|Wv_i\|^2 - \beta}, Wv_i \right) \quad (5) \qquad \bar{v}_i = \left( \frac{(\tilde{v}_i^{\mathbf{Q}_1})^T \sum_{j=1}^N \tilde{v}_j^{\mathbf{K}_1} (\tilde{v}_j^{\mathbf{V}_1})^T}{(\tilde{v}_i^{\mathbf{Q}_1})^T \sum_{j=1}^N \tilde{v}_j^{\mathbf{K}_1}} \right)^T, \quad (7)$$
$$\text{s.t } W \in \{\mathbf{Q}_1, \mathbf{K}_1, \mathbf{V}_1\},$$
$$\tilde{v}_i^{\mathbf{Q}_1}, \tilde{v}_i^{\mathbf{K}_1}, \tilde{v}_i^{\mathbf{V}_1} = \phi(v_{i,[1:]}^{\mathbf{Q}_1}), \phi(v_{i,[1:]}^{\mathbf{K}_1}), v_{i,[1:]}^{\mathbf{V}_1}, \quad (6) \qquad \hat{v}_i = \left( \sqrt{\|\bar{v}_i\|^2 - \beta}, \bar{v}_i \right) \in \mathbb{L}_\beta^{p'}, \quad (8)$$

where $\mathbf{Q}_1, \mathbf{K}_1, \mathbf{V}_1 \in \mathbb{R}^{p' \times (p+1)}$ are three trainable matrices, and $\phi$ is a positive-definite kernel function used to linearize the standard similarity function $\text{sim}(\mathbf{Q}, \mathbf{K})$ in Transformers. In our $\mathcal{Q}$-GT, we choose $\phi(x) = 1 + elu(x)$, which is commonly used in linear attention models (Wu et al., 2022, 2023). The core idea behind hyperbolic linear attention is that each vector $v_i \in \mathbb{L}_\beta^p$ is projected by three matrices $\mathbf{Q}_1, \mathbf{K}_1, \mathbf{V}_1$ in Eq. (5). The kernel function $\phi$ and linear attention are then applied to the space-like components $v_{i,[1:]}$ in Eq. (6) and Eq. (7). Finally, the time-like component is computed to obtain the hidden vector $\hat{v}_i$ in Eq. (8). It is important to note that the hyperbolic linear attention used in this paper differs from Yang et al. (2024), since we do not employ kernel function $\phi$ to the value vector $v_i^{\mathbf{V}_1}$ and omit the residual connection. Similar to hyperbolic linear attention, we introduce spherical linear attention as follows:

$$u_i^W = \left( \sqrt{|\beta|} Wu_i / \|Wu_i\| \right) \quad (9) \qquad \bar{u}_i = \left( \frac{(\tilde{u}_i^{\mathbf{Q}_2})^T \sum_{j=1}^N \tilde{u}_j^{\mathbf{K}_2} (u_j^{\mathbf{V}_2})^T}{(\tilde{u}_i^{\mathbf{Q}_2})^T \sum_{j=1}^N \tilde{u}_j^{\mathbf{K}_2}} \right)^T, \quad (11)$$
$$\text{s.t } W \in \{\mathbf{Q}_2, \mathbf{K}_2, \mathbf{V}_2\},$$
$$\tilde{u}_i^{\mathbf{Q}_2}, \tilde{u}_i^{\mathbf{K}_2} = \phi(u_i^{\mathbf{Q}_2}), \phi(u_i^{\mathbf{K}_2}), \quad (10) \qquad \hat{u}_i = \left( \sqrt{|\beta|} \bar{u}_i / \|\bar{u}_i\| \right) \in \mathbb{S}_{|\beta|}^{q'}, \quad (12)$$

where $\mathbf{Q}_2, \mathbf{K}_2, \mathbf{V}_2 \in \mathbb{R}^{(q'+1) \times (q+1)}$ are three trainable matrices. After applying linear attention, we concatenate the outputs $\hat{u}_i$ and $\hat{v}_i$ to achieve $i^{th}$ node's hidden feature $x_i^{\mathbb{S}_{|\beta|}^{q'} \times \mathbb{L}_\beta^{p'}} = (\hat{u}_i, \hat{v}_i)^T$.

## 4.2 Pseudo-Riemannian refining function

Inspired by Yang et al. (2024), we introduce the product refining function $\sigma$, which generalizes fundamental operations such as dropout, layer normalization and activation function in the product space $\mathbb{S}_{|\beta|}^q \times \mathbb{L}_\beta^p$:

$$\sigma(x) = (\sqrt{|\beta|}\sigma(u)/\|\sigma(u)\|, \sqrt{\|\sigma(v_{[1:]})\|^2 - \beta}, \sigma(v_{[1:]}))^T, \quad (13)$$

where $x = (u, v)^T \in \mathbb{S}_{|\beta|}^q \times \mathbb{L}_\beta^p$ with $u \in \mathbb{S}_{|\beta|}^q$ and $v \in \mathbb{L}_\beta^p$. The product refining function $\sigma$ operates independently on the spherical and hyperbolic manifolds, similar to linear attention, which ensures that the output remains within the product space.

## 4.3 Pseudo-Riemannian residual connection

Residual connection in pseudo-Riemannian methods requires the definition of vector addition in pseudo-Riemannian space. Xiong et al. (2022) suggested using parallel transport with antipodal points as vector addition in the pseudo-hyperboloid $\mathcal{Q}_\beta^{p,q}$. However, this approach alters node features by reversing their original representation, making it unsuitable for preserving feature integrity. Therefore, we define a new vector addition as follows:

**Definition 2** (*Pseudo-hyperboloid vector addition*). *Given two vectors* $x, y \in \mathcal{Q}_\beta^{p,q}$, *the pseudo-hyperboloid vector addition is defined as:*

$$z = y \oplus_\beta x = \exp_{\Phi(y)}^\beta (P_{\bar{\mathbf{o}} \to \Phi(y)}^\gamma (\log_{\bar{\mathbf{o}}}^\beta (\Phi(x)))), \tag{14}$$

*where* $\Phi$ *is the extrinsic mapping function in Eq. (3),* $\exp^\beta$ *and* $\log^\beta$ *are the exponential map and logarithmic map respectively,* $P^\gamma$ *is the parallel transport,* $\bar{\mathbf{o}} = (\sqrt{|\beta|}, 0, ..., 0)$ *is the south pole in the manifold* $\mathcal{Q}_\beta^{p,q+1}$, *and the output* $z \in \mathcal{Q}_\beta^{p,q+1}$.

The procedure for vector addition on the pseudo-hyperboloid is similar to the bias addition in hyperbolic space (Chami et al., 2019). With the extrinsic mapping introduced in Section 3, we enable the application of logarithmic map and parallel transport while avoiding geodesic disconnections and preserving the original features. We further provide the closed form of the vector addition $\oplus_\beta$ in the following proposition:

**Proposition 2.** *Given two vectors* $x, y \in \mathcal{Q}_\beta^{p,q}$, *the closed form of pseudo-hyperboloid vector addition is expressed as:*

$$z = y \oplus_\beta x = \Phi(x) + \frac{\langle \Phi(x), \Phi(y) \rangle_{q+1}}{|\beta|} (\bar{\mathbf{o}} + \Phi(y)).$$

### 4.4 Overall architecture

Fig. 1 illustrates the architecture and data flow of the proposed method $\mathcal{Q}$-GT. During preprocessing, we apply the strategy described in Section 5 to find the appropriate pseudo-hyperboloid for embedding the input graph. Once the manifold is selected, the input features $X^{\mathbb{E}}$ are projected onto $\mathcal{Q}_\beta^{p^*,q^*}$. By using the functions $\Phi$ and $\Psi$ in Eq. (3) and Eq. (4), the projected features are further mapped to a product space $\mathbb{S}_{|\beta|}^{q^*} \times \mathbb{L}_\beta^{p^*}$. The resulting node representations $X^{\mathbb{S}_{|\beta|}^{q^*} \times \mathbb{L}_\beta^{p^*}}$ are processed by the $\mathcal{Q}$-GT module to capture global relations between nodes. Meanwhile, a graph convolutional module $\mathcal{Q}$-GCN2, which is designed based on our diffeomorphic framework, extracts intrinsic graph structural information of the input. Finally, the outputs of $\mathcal{Q}$-GT and $\mathcal{Q}$-GCN2 are aggregated to produce the final graph representation. The details of $\mathcal{Q}$-GCN2 is provided in the Appendix G.

## 5 Space searching algorithm

A pseudo-hyperboloid is defined by two key hyperparameters: the number of time dimensions $q$ and the number of space dimensions $p$. Given a fixed total dimensionality $d = p + q + 1$ for representing node features, varying $p$ and $q$ alters the geometric properties of the pseudo-hyperboloid. Since high-dimensional pseudo-hyperboloids are difficult to visualize directly, their geometric properties can be more easily interpreted by mapping them to the product of spherical and hyperbolic spaces using our diffeomorphic framework. When $p = 0$, the manifold $\mathcal{Q}_\beta^{p,q}$ becomes a sphere $\mathbb{S}_{|\beta|}^q$ which is well-suited for embedding cyclical graphs. As $p$ increases, the topology of $\mathcal{Q}_\beta^{p,q}$ transitions into a product of hyperbolic and spherical geometries, which is appropriate for mixed-structural graphs. When $p$ reaches its maximum value of $d - 1$, $\mathcal{Q}_\beta^{p,q}$ becomes a hyperbolic manifold $\mathbb{L}_\beta^q$ which is effective for modeling tree-like graphs. Once the correlation between the input graph's topology and the space dimension $p$ is known, we can determine the appropriate manifold.

In our work, we rely on the GSC distribution $\mathcal{N}(\bar{\mu}, \bar{\sigma}^2)$ (Bachmann et al., 2020), which provides a comprehensive view of how curvature is distributed across the entire graph, reflecting the geometric complexity of the input graph. We hypothesize that the GSC distribution $\mathcal{N}(\bar{\mu}, \bar{\sigma}^2)$ of input graph is closely related to the topological characteristics of the pseudo-hyperboloid $\mathcal{Q}_\beta^{p,q}$. Therefore, we propose a simple yet effective space searching algorithm to choose a value of $p$ (or $q$) based on the GSC distribution. Specifically, we define a bijective function $\Gamma$ that maps each pseudo-hyperboloid $\mathcal{Q}_\beta^{p_i,q_i}$ to its corresponding ideal GSC distribution $\mathcal{N}(\mu_i, \sigma_i^2)$. To identify the most suitable manifold, we iteratively compute the KL divergence between the input GSC distribution $\mathcal{N}(\bar{\mu}, \bar{\sigma}^2)$ and each ideal GSC distribution. We then select the pseudo-hyperboloid associated with the ideal distribution $\mathcal{N}(\mu_i, \sigma_i^2)$ that yields the smallest KL divergence. This procedure is illustrated in Fig. 1 as the choosing manifold step. Given a collection of pseudo-hyperboloids $\{\mathcal{Q}_\beta^{p_i,q_i}\}_{i=0}^{i=d-1}$ satisfying that

Table 1: The node classification results in F1-score. The best results are in **bold** and the second-best results are in underline for each dataset. OOM denotes out of memory during training or testing.

| Methods | Datasets | | | | | | |
|---|---|---|---|---|---|---|---|
| | Cora | Citeseer | Airport | Pubmed | Arxiv | Penn94 | Twitch Gamers |
| GCN | $81.87 \pm 0.52$ | $70.73 \pm 0.61$ | $74.62 \pm 0.56$ | $79.73 \pm 0.31$ | $57.39 \pm 0.54$ | $80.92 \pm 0.57$ | $63.73 \pm 0.11$ |
| HGCN | $76.57 \pm 0.90$ | $65.60 \pm 0.64$ | $83.46 \pm 0.94$ | $77.09 \pm 0.26$ | $62.16 \pm 0.36$ | $82.04 \pm 0.01$ | $61.65 \pm 0.94$ |
| $\kappa$-GCN | $73.83 \pm 0.56$ | $67.27 \pm 0.71$ | $80.60 \pm 0.39$ | $75.07 \pm 0.58$ | $58.36 \pm 0.18$ | $72.81 \pm 0.17$ | $61.81 \pm 0.72$ |
| $\mathcal{Q}$-GCN | $80.20 \pm 0.41$ | $67.80 \pm 0.51$ | $76.12 \pm 0.48$ | $76.60 \pm 0.54$ | $64.80 \pm 0.22$ | $82.68 \pm 0.14$ | $63.53 \pm 0.30$ |
| $\mathcal{Q}$-GCN2 | $82.60 \pm 0.24$ | $71.63 \pm 0.29$ | $91.66 \pm 0.45$ | $80.13 \pm 0.26$ | $64.93 \pm 0.32$ | $82.60 \pm 0.07$ | $63.77 \pm 0.06$ |
| GraphGPS | $73.76 \pm 1.31$ | $61.07 \pm 0.40$ | $89.31 \pm 0.95$ | $71.53 \pm 0.74$ | OOM | OOM | OOM |
| NodeFormer | $80.27 \pm 0.83$ | $70.03 \pm 0.31$ | $88.87 \pm 0.36$ | **$81.30 \pm 0.65$** | $53.37 \pm 0.34$ | $77.80 \pm 0.11$ | $63.97 \pm 0.17$ |
| SGFormer | $82.00 \pm 0.51$ | **$71.93 \pm 0.59$** | $93.32 \pm 1.09$ | $78.97 \pm 0.09$ | $61.38 \pm 0.25$ | $78.45 \pm 0.19$ | $63.73 \pm 0.10$ |
| Hypformer | $75.13 \pm 0.69$ | $67.77 \pm 0.41$ | $89.19 \pm 0.50$ | $78.93 \pm 0.21$ | $64.62 \pm 0.22$ | $78.13 \pm 0.69$ | $63.16 \pm 0.33$ |
| FPS-T | $60.50 \pm 0.40$ | $60.70 \pm 0.30$ | $91.41 \pm 0.17$ | $76.23 \pm 0.31$ | OOM | OOM | OOM |
| $\mathcal{Q}$-GT | **$82.97 \pm 0.17$** | $70.40 \pm 0.49$ | **$95.70 \pm 0.08$** | $80.37 \pm 0.40$ | **$66.24 \pm 0.05$** | **$82.77 \pm 0.10$** | **$64.45 \pm 0.07$** |

$p_i + q_i = d - 1$, where $d$ is a fixed total dimensionality of embedding space and $d - 1$ is an even number greater than 2, a bijective function $\Gamma : \mathbb{N} \to \mathcal{N}(\mu, \sigma^2)$ is defined as:

$$
\Gamma(p_i) = \begin{cases} \mathcal{N}\left(-|\bar{\mu}|, \left(\sigma_{\max} - (\sigma_{\max} - \sigma_{\min})\left(\frac{2p_i - (d+1)}{d-3}\right)\right)^2\right), & \text{if } (d-1)/2 < p_i \leq d-1 \\ \mathcal{N}(0, 1), & \text{if } p_i = (d-1)/2 \\ \mathcal{N}\left(|\bar{\mu}|, \left(\sigma_{\min} + (\sigma_{\max} - \sigma_{\min})\left(\frac{2p_i}{d-3}\right)\right)^2\right). & \text{if } 0 \leq p_i < (d-1)/2 \end{cases}
$$

$$(15)$$

Since $q_i = d - 1 - p_i$, the function $\Gamma$ can be defined in terms of $p_i$ alone. The mean $\mu_i$ is derived from the practical mean $\bar{\mu}$, while the standard deviation $\sigma_i$ is assumed to vary linearly with the space dimension $p_i$. More details about the space searching algorithm are in Appendix C. In general, the ideal standard deviation may follow a non-linear relationship concerning the space dimension. We leave the exploration of this to future work.

# 6 Experiments

We evaluate the performance of our proposed $\mathcal{Q}$-GT model on node classification and link prediction tasks, and conduct a comprehensive comparison against a wide range of state-of-the-art non-Euclidean GCNs and graph transformers. Our implementation is available at the GitHub repository https://github.com/quanlv9211/QGT.git.

## 6.1 Node classification

**Datasets.** We choose seven widely used graph datasets for evaluation, including four citation networks Cora (McCallum et al., 2000), Citeseer (Giles et al., 1998), Pubmed (Sen et al., 2008), and Arxiv (Hu et al., 2020); two social networks Penn94 and Twitch Gamers (Lim et al., 2021); and one transportation network Airport (Xiong et al., 2022). For the node classification task, we follow the standard dataset splits used by Hu et al. (2020); Xiong et al. (2022); Lim et al. (2021). Summary statistics for all seven datasets are provided in the Appendix I. Notably, the graph sectional curvature (GSC) distributions across most datasets are consistently positive, except Citeseer, where the mean of the GSC distribution is close to 0.

**Baselines.** We compare the performance of $\mathcal{Q}$-GT against state-of-the-art GNNs and graph Transformers operating in both Euclidean and non-Euclidean spaces. The GNN-based baselines include GCN (Kipf and Welling, 2017), HGCN (Chami et al., 2019), $\kappa$-GCN (Bachmann et al., 2020), $\mathcal{Q}$-GCN (Xiong et al., 2022), and our $\mathcal{Q}$-GCN2 which is a GCN variant built upon our proposed diffeomorphic framework. The Transformer-based baselines consist of GraphGPS (Rampášek et al., 2022), NodeFormer (Wu et al., 2022), SGFormer (Wu et al., 2023), Hypformer (Yang et al., 2024), and FPS-T (Cho et al., 2023). To ensure a fair comparison, we set a 16-dimensional hidden embedding for all models. For product-space models such as $\kappa$-GCN and FPS-T, we constrain 2 components on their respective product manifolds. For $\mathcal{Q}$-GT, $\mathcal{Q}$-GCN2, and $\mathcal{Q}$-GCN, we apply the space searching algorithm in Section 5. Each model is trained for up to 1000 epochs with early stopping applied.

**Results.** Table 1 presents the node classification results in terms of F1-score. Our proposed $\mathcal{Q}$-GT consistently outperforms competing methods on 5 out of 7 datasets. The most notable improvement is observed on the Airport dataset, where $\mathcal{Q}$-GT achieves a performance gain of 2.55% in F1-score. On non-homophilous datasets such as Penn94 and Twitch Gamers, where Transformer-based baselines typically underperform, $\mathcal{Q}$-GT demonstrates stable and robust results. For instance, on Penn94, $\mathcal{Q}$-GT surpasses the second-best Transformer (SGFormer) by 5.51% in F1-score. Moreover, by leveraging a linear attention mechanism, $\mathcal{Q}$-GT scales effectively to large graphs (such as Arxiv and Twitch Gamers), while GraphGPS with the standard attention encounters the out-of-memory (OOM) problem. $\mathcal{Q}$-GCN2 also delivers strong performance, ranking second on three datasets (Cora, Citeseer, and Arxiv) and exceeding $\mathcal{Q}$-GCN on most benchmarks. It is worth noting that the Transformer-based baseline FPS-T is similar to $\mathcal{Q}$-GT, as both operate in product spaces of Riemannian manifolds. However, FPS-T suffers from significant computational overhead due to its reliance on tangent mappings and expensive Laplacian eigenvector computations, leading to out-of-memory issues on three large-scale datasets. On the node classification task for Cora and Citeseer, FPS-T underperforms $\mathcal{Q}$-GT, with learned curvatures converging to near-zero values (e.g. $\{0.023, 0.025\}$ on Cora and $\{0.008, 0.007\}$ on Citeseer), which fail to capture the underlying hyperbolic geometry In contrast, $\mathcal{Q}$-GT benefits from operating directly on the product of spherical and hyperbolic manifolds, which is diffeomorphic to the pseudo-hyperboloid, thereby enhancing its performance.

## 6.2 Link prediction

**Setup.** For the link prediction task, we evaluate performance on four datasets: Cora, Citeseer, Pubmed, and Airport. Each dataset's edges are split into $85\%$, $5\%$, and $10\%$ for training, validation, and testing, respectively. The baseline configurations are the same as those in Section 6.1.

**Results.** The results of link prediction are shown in Table 2. $\mathcal{Q}$-GT obtains the highest score on every dataset, with particularly high margins on Cora ($96.87\%$ in AUC) and Citeseer ($97.59\%$ in AUC), outperforming strong transformer baselines like NodeFormer, SGFormer and FPS-T. $\mathcal{Q}$-GCN2 also demonstrates competitive performance, ranking second on Citeseer and Pubmed. Furthermore, both $\mathcal{Q}$-GT and $\mathcal{Q}$-GCN2 significantly surpass $\mathcal{Q}$-GCN across all datasets. For example, $\mathcal{Q}$-GT and $\mathcal{Q}$-GCN2 achieve gains of 43.37 % and 42.81 % in AUC score on the Airport against $\mathcal{Q}$-GCN. These results underscore the effectiveness of our diffeomorphic framework in preserving structural information critical for accurate link prediction.

Table 2: The link prediction results in ROC AUC (%). The best results are in **bold** and the second-best results are in underline for each dataset.

| Methods | Datasets | | | |
| --- | --- | --- | --- | --- |
| | Cora | Citeseer | Airport | Pubmed |
| GCN | $80.75 \pm 0.23$ | $86.12 \pm 0.24$ | $88.92 \pm 0.30$ | $92.62 \pm 0.20$ |
| HGCN | $94.11 \pm 0.18$ | $96.11 \pm 0.11$ | $92.85 \pm 0.29$ | $92.21 \pm 0.08$ |
| $\kappa$-GCN | $94.29 \pm 0.10$ | $95.46 \pm 0.07$ | $85.50 \pm 0.19$ | $87.63 \pm 0.19$ |
| $\mathcal{Q}$-GCN | $94.45 \pm 0.08$ | $96.19 \pm 0.20$ | $67.35 \pm 0.75$ | $85.95 \pm 0.65$ |
| $\mathcal{Q}$-GCN2 | $95.93 \pm 0.07$ | $97.25 \pm 0.05$ | $95.61 \pm 0.05$ | $95.45 \pm 0.19$ |
| GraphGPS | $92.60 \pm 0.17$ | $96.04 \pm 0.12$ | $94.40 \pm 0.04$ | $94.00 \pm 0.10$ |
| NodeFormer | $94.04 \pm 0.13$ | $95.67 \pm 0.02$ | $94.22 \pm 0.09$ | $94.39 \pm 0.08$ |
| SGFormer | $93.70 \pm 0.25$ | $95.74 \pm 0.08$ | $93.96 \pm 0.10$ | $92.10 \pm 0.20$ |
| Hypformer | $95.99 \pm 0.30$ | $96.83 \pm 0.12$ | $96.18 \pm 0.04$ | $94.59 \pm 0.17$ |
| FPS-T | $95.25 \pm 0.12$ | $97.17 \pm 0.44$ | $96.10 \pm 0.13$ | $92.11 \pm 0.04$ |
| $\mathcal{Q}$-GT | **$96.87 \pm 0.19$** | **$97.59 \pm 0.11$** | **$96.56 \pm 0.07$** | **$96.40 \pm 0.10$** |

## 6.3 More ablation studies

**Scalability.** To further assess the scalability of our model, we conducted additional evaluations of $\mathcal{Q}$-GT and other strong baselines on two enormous datasets from the Open Graph Benchmark: Products and Vessels (Hu et al., 2020), where the results are shown in Table 3. This experiment is performed with all Transformer models configured with 1 layer, GNNs with 3 layers, and training conducted over 200 epochs. It is obvious that $\mathcal{Q}$-GT achieves the highest performance on both datasets, outperforming the strongest baselines by 7.3% on Products and 32.09% on Vessel. However, as shown in Appendix H.2 about model complexity, $\mathcal{Q}$-GT introduces higher computational and memory costs

Table 3: Comparison on Open Graph Benchmark datasets. Products is used for the node classification task, while Vessel is used for link prediction.

| Methods | Datasets | |
| --- | --- | --- |
| | Products | Vessel |
| $\mathcal{Q}$-GCN | $29.56 \pm 0.31$ | $59.64 \pm 0.37$ |
| $\mathcal{Q}$-GCN2 | $52.08 \pm 0.45$ | $50.21 \pm 0.36$ |
| SGFormer | $65.47 \pm 0.28$ | $50.03 \pm 0.26$ |
| Hypformer | $40.25 \pm 0.54$ | $49.87 \pm 0.49$ |
| $\mathcal{Q}$-GT | **$70.25 \pm 0.27$** | **$78.78 \pm 0.50$** |

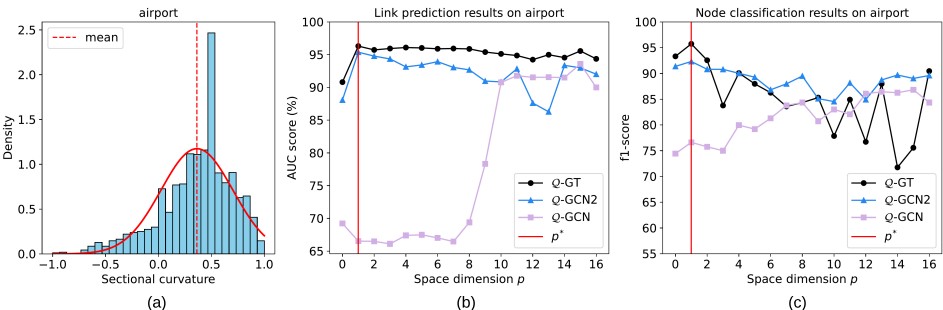

Figure 2: Evaluation of $\mathcal{Q}$-GT, $\mathcal{Q}$-GCN2, and $\mathcal{Q}$-GCN across varying space dimensions on the Airport dataset: (a) GSC distribution. (b) Link prediction performance. (c) Node classification performance. The red line denotes space dimension $p^*$ estimated by the searching algorithm.

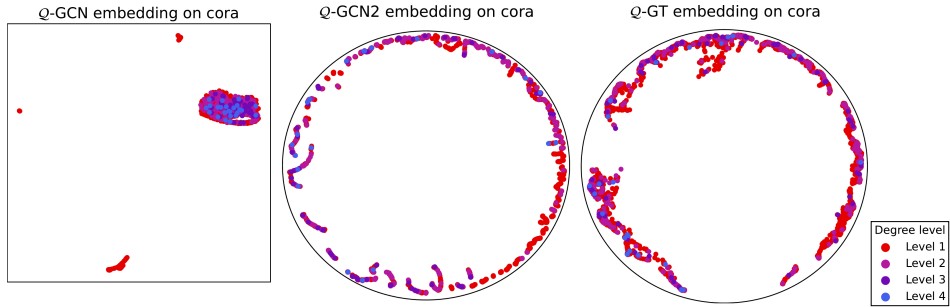

Figure 3: Visualization of the learned embeddings for link prediction task on the Cora dataset, where the colors denote different levels of nodes. The levels are defined according to node degree as follows: Level 1 (1–5), Level 2 (6–10), Level 3 (11–20), and Level 4 (greater than 20).

compared to other baselines. We recognize this as a limitation of our current approach and plan to investigate more efficient solutions in future work.

**Space searching algorithm.** To analyze the effectiveness of space searching algorithm, we conduct an ablation study on the Airport dataset using various space dimensions $p \in [0, 16]$. As shown in Fig. 2, the GSC distribution of Airport network is predominantly positive, indicating that the graph is best embedded in a pseudo-hyperboloid with a low space dimension. Notably, our space searching algorithm correctly identifies $p^* = 1$ as the optimal dimension, aligning with the highest point of performance in $\mathcal{Q}$-GT and $\mathcal{Q}$-GCN2. In contrast, $\mathcal{Q}$-GCN achieves best performance in a large-space dimensional manifold $\mathcal{Q}_\beta^{15,1}$, which is counterintuitive given the structure of the dataset. Nevertheless, $\mathcal{Q}$-GCN still underperforms compared to $\mathcal{Q}$-GT and $\mathcal{Q}$-GCN2 in the optimal manifold. More experimental results of the space searching algorithm are provided in the Appendix H.1.

**Diffeomorphic framework.** Fig. 3 visualizes the learned embedding on the Cora dataset within the product spaces of the estimated $\mathcal{Q}_\beta^{2,14}$. Specifically, we omit the common spherical component used in our diffeomorphism and in the work of Xiong et al. (2022). We instead plot the hyperbolic features for $\mathcal{Q}$-GT, $\mathcal{Q}$-GCN2, and the Euclidean features for $\mathcal{Q}$-GCN using the UMAP tool[1]. We refer to nodes with degree Level 1 as low-level nodes, while all others are considered high-level nodes. In $\mathcal{Q}$-GCN, both high-level and low-level nodes form a single large cluster, meaning that these nodes are oversmoothed in the Euclidean space. In $\mathcal{Q}$-GT and $\mathcal{Q}$-GCN2, high-level nodes are more dispersed, forming multiple small groups that include only a few low-level nodes. These findings suggest that the hyperbolic component in our diffeomorphic framework effectively captures local hierarchical relationships among nodes and separates them into distinct clusters. We further evaluate the effectiveness of the diffeomorphic framework on a toy tree-like graph in the Appendix H.3.

To fully assess our proposed work on downstream tasks, we perform an ablation study in the Appendix H.4 to analyze the impact of each key component in $\mathcal{Q}$-GT.

---

[1]https://umap-learn.readthedocs.io/en/latest/index.html

### 6.4 Further discussions

**Limitations.** The proposed graph models ($\mathcal{Q}$-GT and $\mathcal{Q}$-GCN2), along with the space searching algorithm, are currently limited to the single static graph setting. As a result, they are not directly applicable to graph-level datasets, such as Long Range Graph Benchmark (Dwivedi et al., 2022), or to dynamic graph scenarios. Our space searching approach is based on the assumption of a linear relationship between the ideal standard deviation and the space dimension, which requires further theoretical analysis to validate its correctness. Since $\mathcal{Q}$-GT is built upon the Transformer architecture, it incurs higher computational and memory costs compared to GNNs (see Appendix H.2).

**Broader impacts.** This work represents a theoretical advancement in the field of manifold learning in graphs. There could be some potential societal consequences of our work. For example, our work can be implemented in various graph learning applications in domains such as recommender systems and social networks, which we left for future analysis.

## 7 Conclusion

In this paper, we propose a novel diffeomorphic framework that decomposes pseudo-Riemannian manifolds into the product of spherical and hyperbolic spaces, preserving key geometric properties of pseudo-Riemannian manifolds. Based on our framework, we successfully generalize Graph Transformer to pseudo-Riemannian manifolds, achieving promising results in graph learning tasks. Besides, by analyzing the properties of pseudo-Riemannian manifolds with space dimension, we develop a space searching algorithm to determine an embedding manifold effectively. Experimental results on real-world datasets, especially in the link prediction task, demonstrate that our model outperforms both Riemannian and pseudo-Riemannian baselines. Based on the results, it is possible to extend our framework to other settings such as graph-level learning, dynamic graph learning or in more generalized domains.

## Acknowledgments

This material is based upon work supported by the Air Force Office of Scientific Research under award number FA2386-24-1-4012.

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

# Technical Appendices and Supplementary Material

## A  Notation

Table 4 summarizes all notations mainly used in this work and their corresponding explanations.

Table 4: Main Notations

| Notations | Descriptions |
|---|---|
| $\mathcal{Q}_\beta^{p,q}$ | A pseudo-hyperboloid with $q+1$ time dimensions, $p$ space dimensions, and space curvature $\beta$ |
| $\hat{\mathcal{Q}}_\beta^{p,q}$ | The zero-first submanifold of $\mathcal{Q}_\beta^{p,q}$ |
| $\langle \cdot, \cdot \rangle$ | The inner product in Euclidean space |
| $\langle \cdot, \cdot \rangle_q$ | The inner product in pseudo-Euclidean space |
| $\|\cdot\|$ | The vector norm in Euclidean space |
| $\|\cdot\|_q$ | The vector norm in pseudo-Euclidean space |
| $G = (V_G, E_G)$ | A static graph $G$ with the node set $V_G$ and the edge set $E_G$ |
| $x, y, z, u, v, u', v'$ | A single vector or point |
| $X$ | A set of vectors $x_i$ |
| $N$ | The number of vectors in a vector set $X$ |
| $\mathcal{T}_x\mathcal{M}$ | The tangent space at point $x$ in a manifold $\mathcal{M}$ |
| exp, log | The exponential and logarithmic maps |
| $\mathcal{P}_{x \to y}^\gamma$ | The parallel transport from $\mathcal{T}_x\mathcal{Q}_\beta^{p,q}$ to $\mathcal{T}_y\mathcal{Q}_\beta^{p,q}$ along the geodesic $\gamma$ |
| $D_\gamma(x,y), d_\gamma(x,y)$ | The distance between two points $x, y$ |
| $d$ | The embedding dimension |
| $\mathbb{R}$ | The Euclidean space |
| $\mathbb{N}$ | The natural number set |
| $\mathbb{S}$ | The spherical space |
| $\mathbb{L}$ | The hyperbolic space or the Lorentz model |
| $\Phi$ | The extrinsic mapping |
| $\Psi$ | The diffeomorphic function |
| $\mathcal{N}(\mu, \sigma^2)$ | The Gaussian distribution with the mean $\mu$ and the variance $\sigma^2$ |
| $f$ | The probability density function of the Gaussian distribution |
| $\Gamma$ | The bijective function to Gaussian distributions |

## B  Related works

**Graph embedding in Euclidean space.** Euclidean space has long been the foundation of deep learning because it supports fundamental linear algebraic operations such as vector addition and matrix multiplication. As a result, many early graph representation learning methods were developed within Euclidean geometry and achieved notable successes. One of the pioneering approaches is the Graph Convolutional Network (GCN) introduced by Kipf and Welling (2017), which enabled deep learning-based graph embeddings. To address GCN's limitations in depth and its restricted capacity to capture long-range dependencies, Transformer architecture have been adapted (Vaswani et al., 2017) for graph data. For example, GraphGPS (Rampášek et al., 2022) integrates local message passing with global attention in a hybrid architecture. Despite their superior performance over graph neural networks in small-scale graph applications, Transformer-based methods often struggle with scalability. (Wu et al., 2022, 2023). Standard attention computes pairwise interactions between all node pairs through a softmax function, resulting in quadratic computational complexity. An alternative approach is utilizing linear attention (Katharopoulos et al., 2020), which offers linear computational complexity, for large-scale graph learning. For instance, NodeFormer (Wu et al., 2022) introduces a novel kernelized Gumbel-Softmax message-passing mechanism to approximate the standard attention with linear complexity. SGFormer (Wu et al., 2023) demonstrates the surprising effectiveness of a *single-layer, single-head* linear attention module in node-level tasks. Leveraging

the advantages of linear attention, we propose a pseudo-Riemannian graph model $\mathcal{Q}$-GT built upon the linear Transformer architecture to ensure scalability and efficiency.

**Graph embedding in non-Euclidean space.** As data becomes increasingly complex and diverse, the limitations of Euclidean geometry in graph representation learning are becoming more evident. For instance, Krioukov et al. (2010) proved that hyperbolic geometry is better suited for embedding scale-free and hierarchical graphs due to its exponential expansion property. Consequently, Chami et al. (2019) extended the classic GCN to hyperbolic GCN by shifting the graph operation to the tangent space for executing vector operations. Yang et al. (2024) developed a scalable and robust hyperbolic Transformer called Hypformer, which employs the linear attention mechanism fully on the hyperbolic manifold. For modeling graphs of mixed topologies, $\kappa$-GCN (Bachmann et al., 2020) generalized GCN to the Cartesian product of constant curvature manifolds. However, computing explicit distances in the product of manifolds can introduce bias toward one component over the other, thereby restricting the model's representational capacity (Law, 2021). Alternatively, pseudo-Riemannian geometry, which generalizes spherical and hyperbolic geometries, can address the weakness of product spaces with its intrinsic distance (Law, 2021; Xiong et al., 2022). Law and Stam (2020) proposed the first nonparametric method to learn graph representations with a novel optimization tool on pseudo-Riemannian manifolds. Then Law (2021) designed a graph neural network on the quotient manifold, which is a submanifold of the pseudo-Riemannian manifold to avoid geodesic disconnection. On the other hand, $\mathcal{Q}$-GCN (Xiong et al., 2022) introduced a diffeomorphism that decomposes pseudo-Riemannian manifolds into the product of spherical and Euclidean spaces, enabling a lightweight extension of GCNs to pseudo-Riemannian geometry. Nevertheless, existing methods suffer from three notable limitations: (1) the diffeomorphism to the product of spherical and Euclidean spaces discards hierarchical information inherent in the graph structure; (2) the GNN architecture is prone to the oversquashing problem as the number of layers increases; and (3) previous methods rely on a brute-force algorithm to select the appropriate pseudo-hyperboloid, which is computationally expensive. To address these limitations and attain better outcomes, we propose $\mathcal{Q}$-GT, which is the first Transformer-based graph method in pseudo-Riemannian geometry.

## C   More details about space searching algorithm

### C.1   Gaussian sectional curvature

Sectional curvature is a fundamental concept in Riemannian geometry that describes how a manifold curves in different two-dimensional directions (Lee, 2018). Given a point $x$ on a manifold $\mathcal{M}$ and two linear independent tangent vectors $u, v \in \mathcal{T}_x\mathcal{M}$ spanning a two-dimensional subspace $U$, the sectional curvature measures the Gaussian curvature of the surface $\text{Exp}(U) \subseteq \mathcal{M}$ formed by exponential mapping tangent vectors within $U$. Intuitively, it provides insight into the local geometric properties around a point $x$ (Gu et al., 2018), where positive sectional curvature indicates a locally spherical structure, zero curvature corresponds to a Euclidean-like space, and negative curvature signifies a hyperbolic structure. In the context of graphs, Gu et al. (2018) proposed a discrete curvature analog $\kappa$ to estimate the section curvature in an undirected graph $G$. Given a node $m \in G$, for any two neighbor nodes $b, c$ and a reference node $a \neq m$, the curvature analog $\kappa(m; b, c)$ is defined as:

$$\kappa(m; b, c) = \frac{1}{|V_G| - 1} \sum_{a \in G, a \neq m} \left( \frac{d_G(a, m)}{2} + \frac{d_G^2(b, c)}{8 d_G(a, m)} - \frac{d_G^2(a, b) + d_G^2(a, c)}{4 d_G(a, m)} \right), \quad (16)$$

where $|V_G|$ is the number of nodes in the graph $G$, $d_G(a, b)$ is the length of the shortest path between nodes $a$ and $b$. The sectional curvature $\kappa(m)$ assigned to point $m$ is determined by averaging curvature analogs $\kappa(m; b, c)$ over different neighbors $(b, c)$. In practice, sampling on the neighbors $b, c$ and reference point $a$ is applied to reduce computational overhead (Gu et al., 2018; Bachmann et al., 2020).

To assess the topological structure of the graph $G$, we compute the Gaussian Sectional Curvature (GSC) distribution $\mathcal{N}(\bar{\mu}, \bar{\sigma}^2)$ (Bachmann et al., 2020). In particular, the mean $\bar{\mu}$ and the variance $\bar{\sigma}^2$ of $\mathcal{N}(\bar{\mu}, \bar{\sigma}^2)$ are calculated as:

$$\bar{\mu} = \frac{1}{|V_G|} \sum_{m \in G} \kappa(m) \, , \, \bar{\sigma}^2 = \frac{1}{|V_G|} \sum_{m \in G} (\kappa(m) - \bar{\mu})^2. \quad (17)$$

GSC distribution $\mathcal{N}(\bar{\mu}, \bar{\sigma}^2)$ provides a comprehensive view of how curvature is distributed across the entire graph, reflecting the geometric complexity of the graph. The mean curvature $\bar{\mu}$ captures the fundamental structural properties of the graph (i.e., whether it is predominantly hierarchical or cyclical), while the variance $\bar{\sigma}^2$ measures the degree of topological heterogeneity. A negative mean $\bar{\mu} < 0$ suggests a more hierarchical structure, favoring embedding in a hyperbolic manifold, whereas a positive mean $\bar{\mu} > 0$ indicates a more cyclical structure, which is suited for a spherical manifold. The variance $\bar{\sigma}^2$ further refines this interpretation: a small variance implies a more homogeneous graph topology, closely aligned with either a hyperbolic or spherical space, while a large variance signifies a more diverse topological structure comprising both cyclical and hierarchical parts.

## C.2 Complexity

To ensure the efficiency of our strategy, we begin by uniformly sampling a subset of nodes $\mathcal{A}$ such that $|\mathcal{A}| \ll |V_G|$. For each node $a \in \mathcal{A}$, we precompute the graph distance $d_G(a, b)$ for all node $b \in G$. Consequently, the curvature analog in Eq. (16) is computed as an average over $\mathcal{A}$ when $m \notin \mathcal{A}$ and over $\mathcal{A} \backslash \{m\}$ when $m \in \mathcal{A}$. The time complexity of our algorithm is $O(|\mathcal{A}|(|V_G| + |E_G|) + n_s|\mathcal{A}||V_G| + d)$, where $n_s$ is the number of times we sample two neighbor nodes, $|E_G|$ is the number of edges in graph $G$, and $d$ is the dimension of embedding space. In contrast, a brute-force approach to selecting hyperparameters $p$ and $q$ has a complexity of $O(n_{\text{iter}} d \Delta)$ where $n_{\text{iter}}$ denotes the number of training iterations, and $\Delta$ represents the model complexity. For instance, in a simple GCN (Kipf and Welling, 2017), the model complexity is $O(L(|V_G|d^2 + |E_G|d))$, where $L$ is the number of layers, while in a standard linear Transformer (Katharopoulos et al., 2020), it is $O(|V_G|d^2)$. Since our space searching algorithm scales linearly with $d$, it remains efficient even for high-dimensional embeddings. Meanwhile, the brute-force approach becomes computationally expensive when both $d$ and $n_{\text{iter}}$ are large.

## C.3 Analysis of bijective function $\Gamma$

Constructing the bijective function $\Gamma$ that reflects the relationship between variations in the ideal GSC distribution and the structural characteristics of the pseudo-hyperboloid is a non-trivial task. To facilitate this construction, we analyze the function $\Gamma$ under three different geometric cases, which are balanced, spherical, and hyperbolic. In the balanced case, the mean curvature $\mu_i$ is set as 0 when the input graph exhibits the most uniform structure, containing equal proportions of cyclical and hierarchical topologies. This structure is best embedded in the balanced manifold $\mathcal{Q}_\beta^{p_i,q_i}$ where $p_i = q_i = (d-1)/2$. The standard variance $\sigma_i$ can take any arbitrary value, and we assign it to 1 for simplicity. In the spherical case, when the manifold $\mathcal{Q}_\beta^{p_i,q_i}$ becomes more spherical by setting $p_i < (d-1)/2 < q_i$, it best corresponds to a graph with a positive mean curvature. Therefore, we set $\mu_i = |\bar{\mu}|$, where $\bar{\mu}$ is the estimated mean curvature of input graph. The variance $\sigma_i^2$, defined as in Eq. (15), increases monotonically with increasing $p_i$. This is because as $p_i$ grows, the manifold $\mathcal{Q}_\beta^{p_i,q_i}$ shifts toward a more balanced structure, which is suited for a uniform topological graph with large variance in its GSC distribution. Notably, the standard deviation $\sigma_i$ reaches its minimum value $\sigma_{\text{min}}$ when $p_i = 0$ (a fully spherical manifold), and its maximum value $\sigma_{\text{max}}$ when $p_i = (d-1)/2 - 1$. We determine $\sigma_{\text{max}}$ such that the probability density function at zero curvature, $f(0)$, is nearly equal to $f(\mu_i)$ for the highest variance, while $\sigma_{\text{min}}$ ensures that $f(0) \ll f(\mu_i)$ for the lowest variance. In practice, we assign $\sigma_{\text{min}} = 0.5|\bar{\mu}|$ and $\sigma_{\text{max}} = 5|\bar{\mu}|$. Similarly, we define the GSC distribution $\mathcal{N}(\mu_i, \sigma_i^2)$ for the hyperbolic case in Eq. (15) where $p_i > (d-1)/2 > q_i$. The key difference is that, unlike the spherical case, the mean curvature $\mu_i$ is negative and the standard deviation $\sigma_i$ decreases monotonically as $p_i$ increases.

# D  Geodesic tools of pseudo-hyperboloid

## D.1  Parallel transport

Given two points $x$ and $y$ that are connected by a geodesic $\gamma$ on the manifold $\mathcal{Q}_\beta^{p,q}$, $\xi$ is a vector in the tangent space $\mathcal{T}_x \mathcal{Q}_\beta^{p,q}$, the parallel transport is formulated by Law (2021):

$$P_{x \to y}^\gamma(\xi) = \xi - \frac{\langle \xi, y \rangle_q}{\langle x, y \rangle_q - |\beta|}(y + x). \tag{18}$$

## D.2 Distance

When $x, y \in \mathcal{Q}_\beta^{p,q}$ are geodesic-connected, the induced distance between $x$ and $y$ in pseudo-hyperboloid is given by $d_\gamma(x,y) = \sqrt{\|\log_x(y)\|_q^2}$. However, in cases where $\log_x(y)$ is not defined due to geodesic disconnection, Xiong et al. (2022) proposed an approximation using midpoints. Specifically, the antipodal point $-x$ serves as an intermediary for the broken geodesic between $x$ and $y$, leading to the distance approximation:

$$
D_\gamma(x,y) = \begin{cases} d_\gamma(x,y), & \text{if } \langle x,y \rangle_q < |\beta| \\ d_\gamma(x,-x) + d_\gamma(-x,y) = \pi\sqrt{|\beta|} + d_\gamma(-x,y). & \text{if } \langle x,y \rangle_q \geq |\beta| \end{cases} \tag{19}
$$

## D.3 Exponential and logarithmic maps

The closed form expressions of two tangent maps $\exp_x^\beta(\xi)$ and $\log_x^\beta(y)$ proposed by Law and Stam (2020):

$$
\exp_x^\beta(\xi) = \begin{cases} \cosh\left(\frac{\sqrt{|\langle \xi,\xi \rangle_q|}}{\sqrt{|\beta|}}\right) x + \frac{\sqrt{|\beta|}}{\sqrt{|\langle \xi,\xi \rangle_q|}} \sinh\left(\frac{\sqrt{|\langle \xi,\xi \rangle_q|}}{\sqrt{|\beta|}}\right) \xi, & \text{if } \langle \xi,\xi \rangle_q > 0 \\ x + \xi, & \text{if } \langle \xi,\xi \rangle_q = 0 \\ \cos\left(\frac{\sqrt{|\langle \xi,\xi \rangle_q|}}{\sqrt{|\beta|}}\right) x + \frac{\sqrt{|\beta|}}{\sqrt{|\langle \xi,\xi \rangle_q|}} \sin\left(\frac{\sqrt{|\langle \xi,\xi \rangle_q|}}{\sqrt{|\beta|}}\right) \xi, & \text{if } \langle \xi,\xi \rangle_q < 0 \end{cases} \tag{20}
$$

$$
\log_x^\beta(y) = \begin{cases} \frac{\cosh^{-1}\left(\frac{\langle x,y \rangle_q}{\beta}\right)}{\sqrt{\left(\frac{\langle x,y \rangle_q}{\beta}\right)^2 - 1}} \left(y - \frac{\langle x,y \rangle_q}{\beta} x\right), & \text{if } \frac{\langle x,y \rangle_q}{|\beta|} < -1 \\ y - x, & \text{if } \frac{\langle x,y \rangle_q}{|\beta|} = -1 \\ \frac{\cos^{-1}\left(\frac{\langle x,y \rangle_q}{\beta}\right)}{\sqrt{1 - \left(\frac{\langle x,y \rangle_q}{\beta}\right)^2}} \left(y - \frac{\langle x,y \rangle_q}{\beta} x\right), & \text{if } \frac{\langle x,y \rangle_q}{|\beta|} \in (-1,1) \end{cases} \tag{21}
$$

where $\xi \in \mathcal{T}_x \mathcal{Q}_\beta^{p,q}$ is a tangent vector and $y$ is a point on the manifold $\mathcal{Q}_\beta^{p,q}$.

# E Proof of Theorem 1

**Theorem 1.** *For any point $x \in \hat{\mathcal{Q}}_\beta^{p,q+1}$, there exists a diffeomorphism $\Psi : x \in \hat{\mathcal{Q}}_\beta^{p,q+1} \to \mathbb{S}_{|\beta|}^q \times \mathbb{L}_\beta^p$ that maps $x$ onto the product manifolds of a sphere and a hyperboloid. The mapping and its inverse are given by:*

$$
\Psi(x) = \begin{pmatrix} \sqrt{|\beta|}\frac{u}{\|u\|} \\ \|u\| \\ v \end{pmatrix}, \quad \Psi^{-1}(z) = \begin{pmatrix} 0 \\ \frac{v_0'}{\sqrt{|\beta|}}u' \\ v_{[1:]}' \end{pmatrix}, \tag{22}
$$

*where $x = \begin{pmatrix} 0 \\ u \\ v \end{pmatrix} \in \hat{\mathcal{Q}}_\beta^{p,q+1}$ with $u \in \mathbb{R}^{q+1} \setminus \{\mathbf{0}\}$, $v \in \mathbb{R}^p$, and $z = \begin{pmatrix} u' \\ v' \end{pmatrix}$ with $u' \in \mathbb{S}_{|\beta|}^q$, $v' \in \mathbb{L}_\beta^p$.*

*Proof.* It is easy to show that $\Psi$ and $\Psi^{-1}$ are smooth since their elemental functions $\{\sqrt{|\beta|}\frac{u}{\|u\|}, \|u\|, v, \frac{v_0'}{\sqrt{|\beta|}}u', v_{[1:]}'\}$ are smooth with $u \in \mathbb{R}^{q+1} \setminus \{\mathbf{0}\}$, $v \in \mathbb{R}^p$, $u' \in \mathbb{S}_{|\beta|}^q$, $v' \in \mathbb{L}_\beta^p$. We only need to prove that $\Psi(\Psi^{-1}(z)) = z$ and $\Psi^{-1}(\Psi(x)) = x$.

In the first term $\Psi(\Psi^{-1}(z))$, we have:

$$
\Psi(\Psi^{-1}(z)) = \Psi\left(\begin{pmatrix} 0 \\ \frac{v_0'}{\sqrt{|\beta|}}u' \\ v_{[1:]}' \end{pmatrix}\right) = \begin{pmatrix} \sqrt{|\beta|}\frac{\frac{v_0'}{\sqrt{|\beta|}}u'}{\|\frac{v_0'}{\sqrt{|\beta|}}u'\|} \\ \|\frac{v_0'}{\sqrt{|\beta|}}u'\| \\ v_{[1:]}' \end{pmatrix} = \begin{pmatrix} \sqrt{|\beta|}\frac{u'}{\|u'\|} \\ \frac{v_0'}{\sqrt{|\beta|}}\|u'\| \\ v_{[1:]}' \end{pmatrix}. \tag{23}
$$

Since $u' \in \mathbb{S}_{|\beta|}^q$, we have $\|u'\| = \sqrt{|\beta|}$. Eq. (23) becomes:

$$\Psi(\Psi^{-1}(z)) = \begin{pmatrix} u' \\ v'_0 \\ v'_{[1:]} \end{pmatrix} = \begin{pmatrix} u' \\ v' \end{pmatrix} = z. \tag{24}$$

For the second term $\Psi^{-1}(\Psi(x))$, we have:

$$\Psi^{-1}(\Psi(x)) = \Psi^{-1}\left(\begin{pmatrix} \sqrt{|\beta|}\frac{u}{\|u\|} \\ \|u\| \\ v \end{pmatrix}\right) = \begin{pmatrix} 0 \\ \frac{\|u\|}{\sqrt{|\beta|}}\sqrt{|\beta|}\frac{u}{\|u\|} \\ v \end{pmatrix} = \begin{pmatrix} 0 \\ u \\ v \end{pmatrix} = x \tag{25}$$

$\square$

## F    Proofs of Propositions 1 and 2

**Proposition 1.** *The extrinsic mapping function $\Phi$ and its inverse $\Phi^{-1}$ are smooth, bijective, and isometric.*

*Proof.* Since both the extrinsic function $\Phi$ and its inverse $\Phi^{-1}$ are linear, they are inherently *smooth* mappings.

We also have $\Phi^{-1}(\Phi(x)) = x$ and $\Phi(\Phi^{-1}(y)) = y$, which proves the *bijective* properties of two functions.

To prove the isometric property, let $x, y$ are two points on pseudo-hyperboloid $\mathcal{Q}_\beta^{p,q}$, we have:

$$\langle x, y \rangle_q = \langle \Phi(x), \Phi(y) \rangle_{q+1} \text{ and } \left\|\left(y - \frac{\langle x, y \rangle_q}{\beta}x\right)\right\|_q = \left\|\left(\Phi(y) - \frac{\langle \Phi(x), \Phi(y) \rangle_{q+1}}{\beta}\Phi(x)\right)\right\|_{q+1}.$$

Hence, we have $\|\log_x(y)\|_q = \|\log_{\Phi(x)}(\Phi(y))\|_{q+1}$ or $d_\gamma(x, y) = d_\gamma(\Phi(x), \Phi(y))$ when $x$ and $y$ are geodesic-connected. In the case of geodesic disconnection, it is easy to show that $d_\gamma(-x, y) = d_\gamma(-\Phi(x), \Phi(y))$, leading to the $D_\gamma(x, y) = D_\gamma(\Phi(x), \Phi(y))$ where $D_\gamma$ is the approximated distance in Eq. (19). Therefore, $\Phi$ is *isometric*. The isometric property proof for the inverse function $\Phi^{-1}$ is similar. $\square$

**Proposition 2.** *Given two vectors $x, y \in \mathcal{Q}_\beta^{p,q}$, the closed form of pseudo-hyperboloid vector addition is expressed as:*

$$z = y \oplus_\beta x = \Phi(x) + \frac{\langle \Phi(x), \Phi(y) \rangle_{q+1}}{|\beta|}(\bar{\mathbf{o}} + \Phi(y)).$$

*Proof.* Beginning from the Definition 2, we have the formula of vector addition:

$$z = \exp_{\Phi(y)}^\beta(P_{\bar{\mathbf{o}} \to \Phi(y)}^\gamma(\log_{\bar{\mathbf{o}}}^\beta(\Phi(x)))).$$

Since $\langle \Phi(x), \bar{\mathbf{o}} \rangle_{q+1} = 0$, following the definition of logarithmic map in Eq. (21), we have:

$$\log_{\bar{\mathbf{o}}}^\beta(\Phi(x)) = \frac{\pi}{2}\Phi(x).$$

Applying the parallel transport $P_{\bar{\mathbf{o}} \to \Phi(y)}^\gamma$ in Eq. (18), we have:

$$P_{\bar{\mathbf{o}} \to \Phi(y)}^\gamma\left(\frac{\pi}{2}\Phi(x)\right) = \frac{\pi}{2}\Phi(x) + \frac{\pi}{2}\frac{\langle \Phi(x), \Phi(y) \rangle_{q+1}}{|\beta|}(\bar{\mathbf{o}} + \Phi(y)) = \frac{\pi}{2}\xi.$$

Now we calculate the squared norm of $\xi$:

$$\langle\xi,\xi\rangle_{q+1} = \langle\Phi(x) + \frac{\langle\Phi(x),\Phi(y)\rangle_{q+1}}{|\beta|}(\bar{\mathbf{o}}+\Phi(y)), \Phi(x) + \frac{\langle\Phi(x),\Phi(y)\rangle_{q+1}}{|\beta|}(\bar{\mathbf{o}}+\Phi(y))\rangle_{q+1}$$

$$= -\frac{(\langle\Phi(x),\Phi(y)\rangle_{q+1})^2}{|\beta|} + \langle x + \frac{\langle x,y\rangle_q}{|\beta|}y, x + \frac{\langle x,y\rangle_q}{|\beta|}y\rangle_q$$

$$= -\frac{(\langle x,y\rangle_q)^2}{|\beta|} + \langle x,x\rangle_q + \left(\frac{\langle x,y\rangle_q}{|\beta|}\right)^2\langle y,y\rangle_q + 2\frac{\langle x,y\rangle_q}{|\beta|}\langle x,y\rangle_q$$

$$= -\frac{(\langle x,y\rangle_q)^2}{|\beta|} + \beta + \left(\frac{\langle x,y\rangle_q}{|\beta|}\right)^2\beta + 2\frac{\langle x,y\rangle_q}{|\beta|}\langle x,y\rangle_q$$

$$= -\frac{(\langle x,y\rangle_q)^2}{|\beta|} + \beta - \frac{(\langle x,y\rangle_q)^2}{|\beta|} + 2\frac{(\langle x,y\rangle_q)^2}{|\beta|} = \beta < 0.$$

Since the squared norm $\langle\frac{\pi}{2}\xi, \frac{\pi}{2}\xi\rangle_{q+1} = \left(\frac{\pi}{2}\right)^2\beta < 0$, following the definition of exponential map in Eq. (20), we have:

$$z = \exp_{\Phi(y)}^{\beta}(\frac{\pi}{2}\xi) = \xi.$$

This finishes the proof. $\qquad\square$

## G Pseudo-Riemannian graph convolutional network

This section describes the pseudo-Riemannian graph convolutional network, $\mathcal{Q}$-GCN2, which builds upon the diffeomorphic framework introduced in Section 3. $\mathcal{Q}$-GCN2 comprises three fundamental operations: linear transformation, neighborhood aggregation, and nonlinear activation. As the linear transformation and nonlinear activation have been detailed in Sections 4.1 and 4.2, respectively, we focus here on the formulation of neighborhood aggregation in pseudo-Riemannian geometry. We then present the complete structure of $\mathcal{Q}$-GCN2.

### G.1 Pseudo-Riemannian neighborhood aggregation

Neighborhood aggregation in GCNs (Kipf and Welling, 2017; Chami et al., 2019) relies on scalar multiplication, an operation that has not been defined on the pseudo-hyperboloid $\mathcal{Q}_{\beta}^{p,q}$. To address this, we instead formulate the neighborhood aggregation on the product of spherical and hyperbolic manifolds by using our diffeomorphic framework. Given the neighborhood $\mathcal{N}(i)$ of node $i$ and their features in the product space $\{x_j^{\mathbb{S}_{|\beta|}^q\times\mathbb{L}_\beta^p}\}_{j\in\mathcal{N}(i)}$ where each node vector $x_j^{\mathbb{S}_{|\beta|}^q\times\mathbb{L}_\beta^p} = (u_j\in\mathbb{S}_{|\beta|}^q, v_j\in\mathbb{L}_\beta^p)^T$, and a set of positive weights $\{w_{i,j}\}_{j\in\mathcal{N}(i)}$, we propose the neighborhood aggregation at node $i$ as follows:

$$\tilde{x}_i^{\mathbb{S}_{|\beta|}^q\times\mathbb{L}_\beta^p} = \text{Agg}(\{x_j^{\mathbb{S}_{|\beta|}^q\times\mathbb{L}_\beta^p}\}, \{w_{i,j}\}) = \sqrt{|\beta|}(\frac{\sum_{j\in\mathcal{N}(i)}w_{i,j}u_j}{\|\sum_{j\in\mathcal{N}(i)}w_{i,j}u_j\|}, \frac{\sum_{j\in\mathcal{N}(i)}w_{i,j}v_j}{\sqrt{|\|\sum_{j\in\mathcal{N}(i)}w_{i,j}v_j\|_{\mathbb{L}}^2|}})^T,$$
$$(26)$$

where $\|x\|_{\mathbb{L}}^2 = \langle x,x\rangle_{\mathbb{L}}$ is the Lorentz squared norm. The aggregating function in Eq. (26) is the extension of a hyperbolic centroid formulation proposed by Law et al. (2019). Since the squared distance in the product space can be decomposed to the sum of squared distances in the component space (Gu et al., 2018), i.e. $\|\tilde{x}_i - x_j\|_{\mathbb{S}\times\mathbb{L}}^2 = \|\tilde{x}_i - x_j\|_{\mathbb{S}}^2 + \|\tilde{x}_i - x_j\|_{\mathbb{L}}^2$, we can find the centroid that minimizes the weighted sum of squared distances in the product space by concatenating two centroids in spherical and hyperbolic spaces. In other words, our neighborhood aggregation is a centroid with respect to the squared distance in the product space. We first calculate the spherical centroid from the feature set $\{u_j\}_{j\in\mathcal{N}(i)}$, then combine with the hyperbolic centroid of a set $\{v_j\}_{j\in\mathcal{N}(i)}$.

### G.2 Overall structure

The development of the $\mathcal{Q}$-GCN2 model is a direct consequence of our proposed diffeomorphic framework and the Transformer model $\mathcal{Q}$-GT. With all key operations of $\mathcal{Q}$-GCN2 introduced, we now

summarize the structure of a single $\mathcal{Q}$-GCN2 layer. Given the node features $X^{\mathcal{Q}_\beta^{p,q}} = \{x_i^{\mathcal{Q}_\beta^{p,q}}\}_{i=1}^{i=N}$ of input graph $G$, we follow the procedure in $\mathcal{Q}$-GT to project them onto the product space, resulting in $X^{\mathbb{S}_{|\beta|}^q \times \mathbb{L}_\beta^p} = \Psi(\Phi(X^{\mathcal{Q}_\beta^{p,q}}))$. Next, the representations $X^{\mathbb{S}_{|\beta|}^q \times \mathbb{L}_\beta^p}$ is sent to the message-passing structure in a $\mathcal{Q}$-GCN2 layer:

$$\hat{x}_i^{\mathbb{S}_{|\beta|}^{q'} \times \mathbb{L}_\beta^{p'}} = \left(\hat{u}_i = \left(\sqrt{|\beta|}W_1 u_i / \|W_1 u_i\|\right), \hat{v}_i = \left(\sqrt{\|W_2 v_i\|^2 - \beta}, W_2 v_i\right)\right)^T, \qquad (27)$$

$$\tilde{x}_i^{\mathbb{S}_{|\beta|}^{q'} \times \mathbb{L}_\beta^{p'}} = \text{Agg}(\{\hat{x}_j^{\mathbb{S}_{|\beta|}^{q'} \times \mathbb{L}_\beta^{p'}}\}, \{w_{i,j}\})_{j \in \mathcal{N}(i)}, \qquad (28)$$

$$\bar{x}_i^{\mathbb{S}_{|\beta|}^{q'} \times \mathbb{L}_\beta^{p'}} = \sigma_{\text{Activation}}(\tilde{x}_i^{\mathbb{S}_{|\beta|}^{q'} \times \mathbb{L}_\beta^{p'}}), \qquad (29)$$

where $W_1 \in \mathbb{R}^{(q'+1) \times (q+1)}, W_2 \in \mathbb{R}^{(p') \times (p+1)}$ are two transformation matrices, $w_{i,j} = ((deg(j) + 1)(deg(i) + 1))^{-1/2}$ is the normalized weight from GCN (Kipf and Welling, 2017), and $deg(i)$ is the degree of node $i$. Finally, the output $\bar{X}^{\mathbb{S}_{|\beta|}^{q'} \times \mathbb{L}_\beta^{p'}} = \{\bar{x}_i^{\mathbb{S}_{|\beta|}^{q'} \times \mathbb{L}_\beta^{p'}}\}_{i=1}^{i=N}$ are projected back to the pseudo-Riemannian manifold by using our diffeomorphic framework, i.e. $\bar{X}^{\mathcal{Q}_\beta^{p',q'}} = \Phi^{-1}(\Psi^{-1}(\bar{X}^{\mathbb{S}_{|\beta|}^{q'} \times \mathbb{L}_\beta^{p'}}))$.

### G.3 Integrating into Graph Transformer

Incorporating GCN to Graph Transformer is well-known for graph representation learning (Wu et al., 2023; Yang et al., 2024), which enhances the local structural information. Let $p^*$ and $q^*$ denote the optimal space and time dimensions, estimated by our searching algorithm in Section 5. Given the outputs of $\mathcal{Q}$-GCN2 and $\mathcal{Q}$-GT as $\mathbf{X}_1, \mathbf{X}_2 \in \mathcal{Q}_\beta^{p^*,q^*}$, we utilize this approach and combine two outputs with a balance hyperparameter $\alpha$ as follows:

$$\begin{aligned}
\bar{\mathbf{X}}_1 &= \log_{\mathbf{o}}^\beta(\Phi(\mathbf{X}_1)), \bar{\mathbf{X}}_2 = \log_{\mathbf{o}}^\beta(\Phi(\mathbf{X}_2)) \\
\bar{Z} &= \alpha \bar{\mathbf{X}}_1 + (1 - \alpha)\bar{\mathbf{X}}_2 \\
Z &= \exp_{\mathbf{o}}^\beta(\bar{Z}) \in \hat{\mathcal{Q}}_\beta^{p^*,q^*+1}
\end{aligned} \qquad (30)$$

where $\Phi$ is the extrinsic mapping function in Eq. (3), $\hat{\mathcal{Q}}_\beta^{p^*,q^*+1}$ is the zero-first submanifold, $\exp_{\mathbf{o}}^\beta$ and $\log_{\mathbf{o}}^\beta$ are the exponential and logarithmic maps in Eq. (20) and Eq. (21). In practice, the value of the balance weight $\alpha$ is determined by the grid search strategy.

## H More ablation studies

### H.1 Effectiveness of space searching algorithm

We further assess the effectiveness of the proposed space searching algorithm on Cora, Citeseer and Pubmed datasets, as visualized in Fig. 4. It is clearly seen that varying space dimensions significantly impact the performance of pseudo-Riemannian models, especially in node classification task. This highlights the importance of selecting an appropriate pseudo-hyperboloid for effective graph representation learning. Our algorithm consistently identifies an optimal or near-optimal space dimension $p^*$ in both tasks. Meanwhile, the performance of $\mathcal{Q}$-GCN does not align with the space searching result or the underlying topology of the input graph. While $\mathcal{Q}$-GCN performs reasonably well when the product space becomes Euclidean space through increasing space dimension, Euclidean space lacks the capacity to capture the cyclical and hierarchical structures in many real-world graphs. This limitation leads to inferior performance compared to our proposed $\mathcal{Q}$-GCN2 and $\mathcal{Q}$-GT.

### H.2 Complexity comparison

For a better complexity comparison, we report the training time and memory of our proposed $\mathcal{Q}$-GCN2 and $\mathcal{Q}$-GT against some strong baselines in Table 5 and Table 6. A comparison between $\mathcal{Q}$-GCN and $\mathcal{Q}$-GCN2 shows that our diffeomorphic framework achieves better efficiency in both runtime and memory. This is because $\mathcal{Q}$-GCN framework relies on a combination of diffeomorphic and tangent space mappings in product manifolds, whereas our approach operates directly within the product manifold space. Nonetheless, $\mathcal{Q}$-GT incurs higher computational and memory overhead

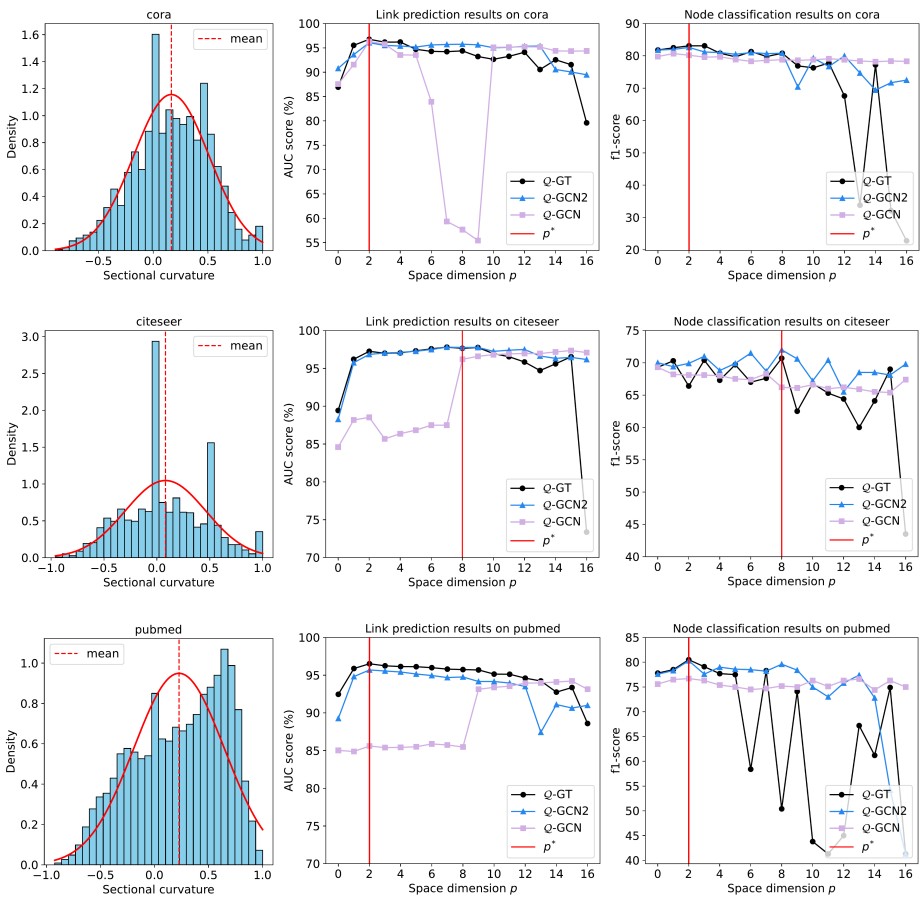

Figure 4: Evaluating space searching algorithm on Cora, Citeseer and Pubmed datasets. The red line denotes space dimension $p^*$ estimated by the searching algorithm.

Table 5: Comparison of training time and the runtime of the space searching algorithm (in second).

| Datasets | SGFormer | Hypformer | $\mathcal{Q}$-GCN | $\mathcal{Q}$-GCN2 | $\mathcal{Q}$-GT | Space searching | Ratio to $\mathcal{Q}$-GT |
|---|---|---|---|---|---|---|---|
| Cora | 40.92 | 66.42 | 136.6 | 82.44 | 193.7 | 12.70 | 0.07x |
| Citeseer | 40.21 | 67.31 | 141.8 | 76.12 | 188.2 | 6.90 | 0.04x |
| Airport | 71.55 | 97.20 | 211.4 | 66.59 | 249.5 | 15.30 | 0.06x |
| Pubmed | 144.8 | 153.8 | 278.0 | 83.52 | 386.0 | 71.60 | 0.19x |
| Arxiv | 100.1 | 136.7 | 347.2 | 145.9 | 353.5 | 1922.1 | 5.43x |
| Penn94 | 65.49 | 140.1 | 299.7 | 121.4 | 258.5 | 361.1 | 1.40x |
| Twitch Gamers | 163.3 | 148.7 | 377.2 | 181.2 | 386.3 | 1611.7 | 4.18x |
| Products | 354.1 | 371.8 | 461.5 | 325.2 | 681.2 | 4081.0 | 5.99x |
| Vessel | 994.5 | 931.4 | 1137.2 | 664.0 | 1024.8 | 1051.3 | 1.02x |
| Avg Rank | 1.6 | 2.1 | 3.8 | 1.7 | 4.2 | - | - |

compared to GNNs, SGFormer, and HypFormer, particularly on large-scale datasets like Products and Vessel. We attribute this overhead to Transformer components such as linear attention, the refining function, and residual connections, which we recognize as a limitation of our method. However, this trade-off enables $\mathcal{Q}$-GT to deliver strong performance on large datasets. We are optimistic that future work can further reduce the model's complexity and retain its effectiveness.

In addition, Table 5 presents a comparison between the runtime of the space searching algorithm and the training time of $\mathcal{Q}$-GT across 9 datasets. On Cora, Citeseer, Pubmed, and Airport, the computing time searching algorithm is less than 20% training time of $\mathcal{Q}$-GT. However, in the two large datasets, Arvix, Twitch Gamers and Products, the space searching requires more computation, around 4-6

Table 6: Comparison of GPU memory (in MiB).

| Datasets | SGFormer | Hypformer | $\mathcal{Q}$-GCN | $\mathcal{Q}$-GCN2 | $\mathcal{Q}$-GT |
|---|---|---|---|---|---|
| Cora | 58.4 | 158.6 | 111.7 | 65.9 | 70.8 |
| Citeseer | 126.8 | 471.2 | 314.8 | 134.5 | 129.6 |
| Airport | 42.8 | 57.9 | 107.3 | 58.4 | 82.0 |
| Pubmed | 195.5 | 382.9 | 428.2 | 240.2 | 348.6 |
| Arxiv | 1290.1 | 3822.7 | 4536.0 | 1106.2 | 3098.1 |
| Penn94 | 2246.6 | 7281.5 | 4876.2 | 2319.9 | 1585.1 |
| Twitch Gamers | 1638.4 | 1578.0 | 2100.2 | 2840.0 | 2713.5 |
| Products | 27979.5 | 32363.3 | 58954.8 | 34668.1 | 51574.0 |
| Vessel | 8016.3 | 17376.8 | 48672.1 | 14249.9 | 39293.8 |
| Avg Rank | 1.1 | 3.4 | 4.4 | 2.7 | 2.7 |

Table 8: Ablation study on the impact of each component in $\mathcal{Q}$-GT.

| Methods | Datasets | | | |
|---|---|---|---|---|
| | Airport | Pubmed | Tree-2 | Tree-3 |
| $\mathcal{Q}$-GT | $96.56 \pm 0.07$ | $96.40 \pm 0.10$ | $94.02 \pm 0.06$ | $92.03 \pm 0.11$ |
| **w/o residual** | $95.22 \pm 0.12$ | $96.24 \pm 0.15$ | $89.66 \pm 2.14$ | $90.70 \pm 0.38$ |
| *Gain(%)* | - 1.39 | - 0.17 | - 4.64 | - 1.45 |
| **w/o refining** | $95.73 \pm 0.07$ | $95.33 \pm 0.28$ | $81.43 \pm 0.63$ | $89.98 \pm 0.27$ |
| *Gain(%)* | - 0.86 | - 1.11 | - 13.4 | - 2.23 |
| **w/o GCN** | $94.43 \pm 0.35$ | $90.47 \pm 0.18$ | $86.78 \pm 0.49$ | $83.53 \pm 1.86$ |
| *Gain(%)* | - 2.21 | - 6.15 | - 7.70 | - 9.24 |

times higher than the training of $\mathcal{Q}$-GT. Compared to a brute-force approach which requires training the model 16 times (once per each dimension), our algorithm is more efficient in both time and computation.

## H.3 Advantage of diffeomorphic framework

To investigate the advantage of the proposed diffeomorphic framework, we create three datasets Tree-1, Tree-2 and Tree-3 that best reflect the hyperbolic geometry. Detailed descriptions of these datasets are provided in Appendix I.1. Table 7 presents the link prediction results of $\mathcal{Q}$-GCN, $\mathcal{Q}$-GCN2 and $\mathcal{Q}$-GT on these graphs. $\mathcal{Q}$-GCN2 and $\mathcal{Q}$-GT significantly outperform $\mathcal{Q}$-GCN across all datasets. For instance, on Tree-2, $\mathcal{Q}$-GT and $\mathcal{Q}$-GCN2 achieve

Table 7: The link prediction results in ROC AUC (%) on tree datasets.

| Datasets | Tree-1 | Tree-2 | Tree-3 |
|---|---|---|---|
| Space dim $p^*$ | 16 | 15 | 13 |
| Time dim $q^*$ | 0 | 1 | 3 |
| $\mathcal{Q}$-GCN | $91.44 \pm 0.52$ | $85.83 \pm 0.51$ | $84.86 \pm 0.35$ |
| $\mathcal{Q}$-GCN2 | $95.34 \pm 0.31$ | $93.60 \pm 0.32$ | $89.25 \pm 0.34$ |
| $\mathcal{Q}$-GT | $96.05 \pm 0.10$ | $94.02 \pm 0.06$ | $92.03 \pm 0.11$ |

AUC improvements of 9.54 % and 9.05 %, respectively, over $\mathcal{Q}$-GCN. To shed light on the superior performances of our models, we further visualize the learned embeddings on the Tree-1, Tree-2, and Tree-3 datasets in Fig. 5. Following the experimental setup in Section 6.3, the embeddings are projected to the product spaces, excluding the spherical component for clearer interpretation. In $\mathcal{Q}$-GCN, low-level nodes are scattered randomly among high-level nodes. $\mathcal{Q}$-GCN2 and $\mathcal{Q}$-GT, by contrast, exhibit a clear clustering of low-level nodes, indicating that these models effectively distinguish between low-level and high-level nodes. These observations emphasize the advantage of our diffeomorphic framework for preserving hierarchical structures of graphs.

## H.4 Impact of key factors in $\mathcal{Q}$-GT

To analyze the impact of each key component in our proposed model, we perform an ablation study using three variants derived from $\mathcal{Q}$-GT:

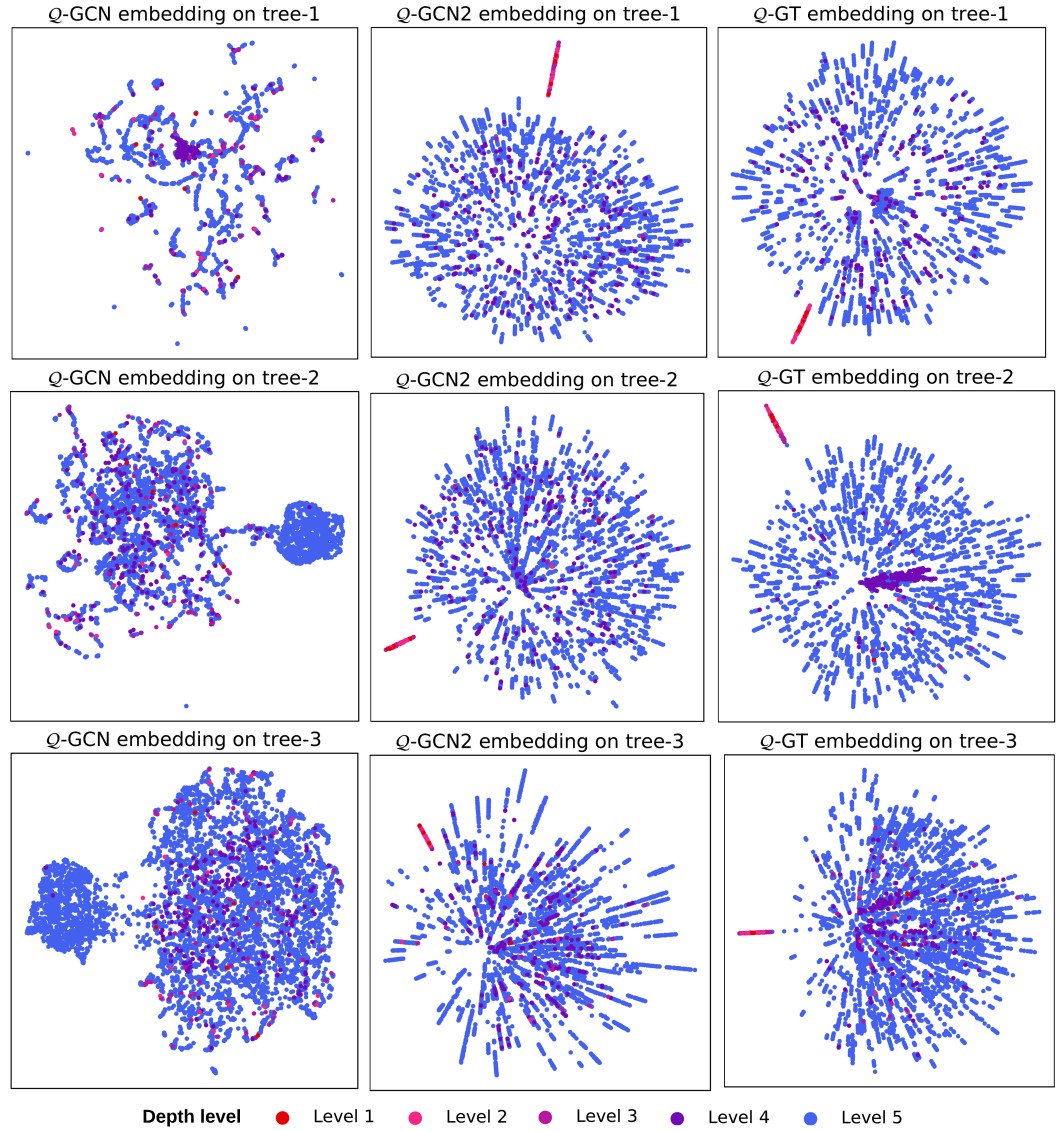

Figure 5: Visualization of the learned embeddings for the link prediction task on Tree-1, Tree-2, Tree-3 datasets. The colors denote depth levels. Nodes at Levels 1 and 2 are classified as low-level, which are closer to the root, while the other are considered high-level.

- **w/o residual**: $\mathcal{Q}$-GT without residual connections. All residual connections are removed from each layer of the model.

- **w/o refining**: $\mathcal{Q}$-GT without refinement operations, including layer normalization, dropout, and non-linear activation. This variant consists solely of linear layers.

- **w/o GCN**: A variant of $\mathcal{Q}$-GT that excludes the graph-based module $\mathcal{Q}$-GCN2, relying solely on the Transformer module for representation learning.

We use the same experimental setup as $\mathcal{Q}$-GT to evaluate these variants, and the link prediction results are reported in Table 8. In general, removing any component would lower the model's performance. **w/o residual** leads to a decline in the model's performance by 1.91% in average. This suggests that residual connections play an important role in stabilizing training and facilitating gradient flow across layers, which is particularly beneficial in deep architectures like $\mathcal{Q}$-GT. Similarly, in the case of **w/o refining**, the model's performance also drops dramatically on Tree-2, which confirms the necessity of refining functions, such as normalization, dropout, and non-linear activation, for effective graph

Table 9: Link prediction results with different embedding dimensions.

| Methods | Datasets | | | |
|---|---|---|---|---|
| | Airport | Pubmed | Tree-2 | Tree-3 |
| Original $\mathcal{Q}$-GT (**16**) | $96.56 \pm 0.07$ | $96.40 \pm 0.10$ | $94.02 \pm 0.06$ | $92.03 \pm 0.11$ |
| $\mathcal{Q}$-GT (**8**) | $95.36 \pm 0.28$ | $95.88 \pm 0.20$ | $87.83 \pm 1.35$ | $85.50 \pm 3.64$ |
| *Gain(%)* | - 1.24 | - 0.54 | - 6.58 | - 7.10 |
| $\mathcal{Q}$-GT (**32**) | $96.09 \pm 0.15$ | $96.48 \pm 0.10$ | $94.36 \pm 0.36$ | $90.29 \pm 2.08$ |
| *Gain(%)* | - 0.49 | + 0.08 | + 0.36 | - 1.89 |

learning in pseudo-Riemannian spaces. When removing graph-based module $\mathcal{Q}$-GCN2, it causes a significant decrease in the performance of $\mathcal{Q}$-GT on all datasets. This observation reveals that the Transformer module alone is insufficient for capturing local structural information in the graph.

Additionally, we evaluate the capacity of $\mathcal{Q}$-GT with different embedding dimensions, 8 and 32, in Table 9. Reducing the dimension to 8 leads to a notable decrease in the link prediction results on hierarchical datasets, whereas increasing to 32 offers only minor improvements. Therefore, we recommend using 16 dimensions to maintain computational efficiency and obtain strong performances.

# I  Experimental details

Table 10: Summary statistics of the real-world datasets.

| Datasets | Cora | Citeseer | Airport | Pubmed | Arxiv | Penn94 | Twitch Gamers | Products | Vessel |
|---|---|---|---|---|---|---|---|---|---|
| **Total nodes** | 2708 | 3327 | 3188 | 19717 | 169343 | 41554 | 168114 | 2449029 | 3538495 |
| **Total edges** | 5278 | 4552 | 18630 | 44324 | 1166234 | 1362229 | 6797557 | 61859140 | 5345897 |
| **Undirected** | Yes | Yes | Yes | Yes | No | Yes | Yes | Yes | Yes |
| **Total classes** | 7 | 6 | 4 | 3 | 40 | 2 | 2 | 47 | - |
| **Features** | 1433 | 3703 | 11 | 500 | 128 | 4814 | 7 | 100 | 3 |
| **Mean** $\bar{\mu}$ | 0.1658 | 0.0799 | 0.3626 | 0.2264 | 0.2290 | 0.2340 | 0.3238 | 0.2550 | -0.1160 |
| **Std** $\bar{\sigma}$ | 0.2290 | 0.3716 | 0.3180 | 0.4197 | 0.3030 | 0.2050 | 0.2088 | 0.2562 | 0.2810 |
| **Space dim** $p*$ | 2 | 8 | 1 | 2 | 1 | 1 | 0 | 1 | 9 |
| **Time dim** $q*$ | 14 | 8 | 15 | 14 | 15 | 15 | 16 | 15 | 7 |

Table 11: Summary statistics of tree datasets.

| Datasets | Tree-1 | Tree-2 | Tree-3 |
|---|---|---|---|
| **Total nodes** | 5461 | 5461 | 5461 |
| **Total edges** | 5460 | 5960 | 6460 |
| **Undirected** | Yes | Yes | Yes |
| **Total classes** | - | - | - |
| **Features** | 5461 | 5461 | 5461 |
| **Mean** $\bar{\mu}$ | -0.5983 | -0.2924 | -0.1417 |
| **Std** $\bar{\sigma}$ | 0.1082 | 0.2953 | 0.2932 |
| **Space dim** $p*$ | 16 | 15 | 13 |
| **Time dim** $q*$ | 0 | 1 | 3 |

## I.1  Datasets

Table 10 presents the statistics of all datasets used in this study. The mean $\bar{\mu}$ and standard deviation $\bar{\sigma}$ denote the Gaussian sectional curvature (GSC) distribution of each input graph. Among all real-world datasets, eight graphs exhibit positive curvature distributions, while other four networks have negative distributions. In terms of graph size, Arxiv, Twitch Gamers, Products and Vessel are large-scale networks, Pubmed and Penn94 are medium-sized, and the remaining datasets are considered small. Detailed descriptions of each dataset are provided as follows:

- **Cora** (McCallum et al., 2000) is a citation network comprising 2708 scientific publications each node is classified into one of seven classes. The network includes 5278 citation links among these papers. Each node is represented by a binary word vector, where each entry indicates the presence or absence of a specific word from a dictionary of 1433 unique terms.

- **Citeseer** (Giles et al., 1998) is a citation network consisting of 3327 scientific publications classified into one of six classes. The citation network consists of 4552 links. Each publication is described by a binary word vector showing the absence/presence of the corresponding word from the dictionary of 3703 unique terms.

- **Pubmed** (Sen et al., 2008) comprises 19717 scientific articles from the PubMed database related to diabetes, each categorized into one of three classes. The citation network includes 44324 links between these publications. Each article is represented by a TF-IDF weighted word vector, constructed from a vocabulary of 500 unique words.

- **Airport** (Xiong et al., 2022) is a transportation network containing 3188 airports. Edges correspond to airline routes connecting the airports. For the node classification task, each airport is labeled with the population of the country in which it is located.

- **Arxiv** (Hu et al., 2020) is a directed citation network between all Computer Science (CS) arXiv papers. This dataset belongs to the Open Graph Benchmark (OGB) collection. Each node is an arXiv paper and each directed edge indicates that one paper cites another one. Each paper comes with a 128-dimensional feature vector obtained by averaging the embeddings of words in its title and abstract. We apply the space searching algorithm on the undirected structure of Arxiv network, while training on the original directed graph.

- **Penn94** (Lim et al., 2021) is a Facebook "friendship" networks at one hundred American colleges and universities from 2005. In this graph, nodes correspond to individual students, each labeled by their reported gender. Node features include academic major, second major or minor, dormitory or housing assignment, graduation year, and high school background.

- **Twitch Gamers** (Lim et al., 2021) is an undirected graph representing connections between user accounts on the Twitch streaming platform. Nodes correspond to individual Twitch accounts, with edges indicating mutual follower relationships. Node features include the number of views, account creation and update timestamps, language, account lifetime, and a flag indicating whether the account is inactive. The associated binary classification task involves predicting whether a channel contains explicit content.

- **Products** (Hu et al., 2020) is an undirected, unweighted graph representing the Amazon co-purchasing network. Each node corresponds to a product on Amazon, and an edge between two nodes indicates that the products were frequently bought together. Node features are derived from the bag-of-words representations of product descriptions, followed by dimensionality reduction using Principal Component Analysis (PCA) to obtain 100-dimensional feature vectors.

- **Vessel** (Hu et al., 2020) is an undirected, unweighted spatial graph capturing the entire mouse brain. Each node is represented by 3D spatial coordinates (x, y, z) within the Allen Brain Atlas reference framework. This dataset is designed to encourage neuroscience researchers to adopt graph-based representations in their studies. At the same time, it presents machine learning researchers with challenging problems, such as integrating biological priors into learning models and scaling algorithms to efficiently process sparse, spatial graphs containing millions of nodes and edges.

- **Tree datasets** are created due to the lack of the negative curvature graphs. Tree-1 is a balanced tree of depth 6 and branching factor 4 consisting of 5461 nodes and 5460 edges. Tree-2 and Tree-3 are derived from Tree-1 by randomly adding 500 and 1000 edge respectively, introducing different magnitudes of curvature. Each node is initialized with a one-hot feature vector, and no node labels are provided in these datasets.

## I.2 Implementation setup

**Objective function.** In the node classification task, following Xiong et al. (2022), we project the output of the final layer of both $\mathcal{Q}$-GT and $\mathcal{Q}$-GCN2 onto the tangent space and apply Euclidean multinomial logistic regression for classification. For link prediction, non-Euclidean methods utilize

the Fermi-Dirac decoder (Krioukov et al., 2010), which computes edge probability scores based on geodesic distances in the embedding space.

**Optimization.** The trainable parameters of $\mathcal{Q}$-GT and $\mathcal{Q}$-GCN2 are defined in Euclidean space via the proposed diffeomorphic framework. As a result, standard Euclidean optimization algorithms, such as Adam (Kingma and Ba, 2015) and SGD (Robbins and Monro, 1951), can be directly applied for model training. RiemannianAdam (Becigneul and Ganea, 2019) is another choice for non-Euclidean methods such as HGCN and Hypformer, which operate under specific manifold constraints.

**Setting.** We implement our models by Pytorch and Geoopt tool [2]. All the experiments are conducted on a GPU device NVIDIA GeForce RTX 4090 with 24GB memory, except for the Products and Vessel datasets, which are run on an NVIDIA A100 GPU with 80GB of memory. For comparison baselines, we follow the implementation of each model introduced in the corresponding paper. The experiment results are reported by the average value with standard deviation on the test sets using three different seeds. Unlike the work of Xiong et al. (2022), we utilize initial features provided in the datasets for node classification instead of pretrained embeddings for evaluating model's learning capacity.

Table 12: The grid search ranges for hyperparameters.

| Hyperparameter | Search range |
|---|---|
| Learning rate $lr$ | 5e-2, 2e-2, 1e-2, 5e-3, 2e-3, 1e-3 |
| Balance weight $\alpha$ | 0.5, 0.6, 0.7, 0.8, 0.9 |
| Number of layers | 1,2,3,4 |
| Activation | relu, tanh, sigmoid, leakyrelu |
| Dropout rate | 0.0, 0.1, 0.2, 0.3, 0.4, 0.5, 0.6, 0.7 |
| Weight decay | 1e-1, 5e-2, 1e-2, 5e-3, 1e-3, 5e-4, 1e-4, 5e-7 |

**Hyperparameter.** Besides the space and time dimensions of pseudo-hyperboloids $\mathcal{Q}_\beta^{p,q}$, our models involves other hyperparameters, including learning rate $lr$, balance weight $\alpha$, number of layers, activation function, dropout rate, and weight decay. The optimal hyperparameters are obtained by applying grid search strategy, where the ranges summarized in Table 12.

### I.3 Pseudocode

In this section, we provide the pseudocode of our proposed framework to facilitate implementation for the reader. Algorithm 1 outlines the procedure of the space searching algorithm. In line 1, a subset $\mathcal{A}$ is uniformly sampled from the node set $V_G$. Lines 2–6 precompute the shortest path distances from each node $a \in \mathcal{A}$ to all other nodes in the graph $G$. The Gaussian sectional curvature (GSC) distribution of the input graph is computed in lines 7–14. Finally, in lines 15–20, we iteratively calculate the KL divergence between the input GSC distribution $\mathcal{N}(\bar{\mu}, \bar{\sigma}^2)$ and each ideal distribution $\mathcal{N}(\mu_i, \sigma_i^2)$ to determine the optimal space dimension $p_i$.

Algorithm 2 demonstrates the learning process of $\mathcal{Q}$-GCN2. In line 1, we apply Algorithm 1 to select the embedding manifold $\mathcal{Q}_\beta^{p^*,q^*}$. Line 2 projects the initial features to the embedding manifold with a function $f$ which is either a dense layer or a graph convolution layer to ensure the dimensional consistency. The projected representation is passed through $L$ layers of $\mathcal{Q}$-GCN2 in lines 3-9, where each layer comprises feature transformation, neighborhood aggregation, and nonlinear activation.

The learning process of $\mathcal{Q}$-GCT is described in Algorithm 3. Similar to $\mathcal{Q}$-GCN2, lines 1–2 involve selecting the appropriate manifold and transforming node features. The transformed representation is then processed through an $L$-layer $\mathcal{Q}$-GT in lines 3-15. To capture local structural information, $\mathcal{Q}$-GCN2 is incorporated in line 16. The outputs of $\mathcal{Q}$-GCN2 and $\mathcal{Q}$-GCT are combined using a balance weight $\alpha$ in lines 17-20. The final node representation $Z$ is obtained for downstream tasks.

---

[2]https://github.com/geoopt/geoopt

---

**Algorithm 1** The procedure of space searching algorithm

---

**Input**: An input graph $G = (V_G, E_G)$

**Output**: The optimal pseudo-hyperboloid $\mathcal{Q}_\beta^{p^*, q^*}$

1: Uniformly sampling a subset of nodes $\mathcal{A} \subset V_G$
2: **for** each node $a \in \mathcal{A}$ **do**
3:     **for** each node $b \in V_G$ **do**
4:         Precomputing the shortest path distance $d_G(a, b)$
5:     **end for**
6: **end for**
7: **for** each node $m \in V_G$ **do**
8:     **for** $i = 1$ to $n_s$ **do**
9:         Uniformly sampling two neighbor nodes $b, c \in \mathcal{N}(m)$
10:         Calculating the curvature analog $\kappa(m; b, c)$ in Eq. (16)
11:     **end for**
12:     Calculating the sectional curvature $\kappa(m)$ by averaging curvature analogs $\kappa(m; b, c)$
13: **end for**
14: Calculating the mean $\bar{\mu}$ and the variance $\bar{\sigma}^2$ in Eq. (17)
15: **for** $i = 0$ to $d - 1$ **do**
16:     Assigning space dimension $p_i := i$
17:     Calculating the ideal GSC distribution $\mathcal{N}(\mu_i, \sigma_i^2) = \Gamma(p_i)$
18:     Calculating the KL divergence $d_{KL}(\mathcal{N}(\bar{\mu}, \bar{\sigma}^2), \mathcal{N}(\mu_i, \sigma_i^2))$
19:     Saving $p^* = p_i$ that yields the smallest KL divergence
20: **end for**
21: **return** the optimal space dimension $p^*$

---

**Algorithm 2** $L$-layer $\mathcal{Q}$-GCN2 learning process

---

**Input**: An input graph $G = (V_G, E_G)$ with the initial features $X^{\mathbb{E}}$

**Output**: A hidden representation $X_L^{\mathcal{Q}_\beta^{p^*, q^*}}$

1: Determining the embedding manifold $\mathcal{Q}_\beta^{p^*, q^*}$ by using Algorithm 1

2: Projecting initial features $X_0^{\mathcal{Q}_\beta^{p^*, q^*}} = \exp_{\mathbf{o}}^\beta([0 || f(X^{\mathbb{E}})])$

3: **for** $l = 1$ to $L$ **do**

4:     Decomposing features $X_{l-1}^{\mathbb{S}_{|\beta|}^{q^*} \times \mathbb{L}_\beta^{p^*}} = \Psi(\Phi(X_{l-1}^{\mathcal{Q}_\beta^{p^*, q^*}}))$

5:     $\hat{X}_{l-1}^{\mathbb{S}_{|\beta|}^{q^*} \times \mathbb{L}_\beta^{p^*}} = \left( \sqrt{|\beta|} W_1 X_{l-1}^{\mathbb{S}_{|\beta|}^{q^*}} / \|W_1 X_{l-1}^{\mathbb{S}_{|\beta|}^{q^*}}\|, \sqrt{\|W_2 X_{l-1}^{\mathbb{L}_\beta^{p^*}}\|^2 - \beta}, W_2 X_{l-1}^{\mathbb{L}_\beta^{p^*}} \right)^T$

6:     $\tilde{X}_{l-1}^{\mathbb{S}_{|\beta|}^{q^*} \times \mathbb{L}_\beta^{p^*}} = \mathrm{Agg}(\hat{X}_{l-1}^{\mathbb{S}_{|\beta|}^{q^*} \times \mathbb{L}_\beta^{p^*}}, E_G)$

7:     $\bar{X}_{l-1}^{\mathbb{S}_{|\beta|}^{q^*} \times \mathbb{L}_\beta^{p^*}} = \sigma_{\mathrm{Activation}}(\tilde{X}_{l-1}^{\mathbb{S}_{|\beta|}^{q^*} \times \mathbb{L}_\beta^{p^*}})$

8:     Projecting back to the pseudo-hyperboloid $X_l^{\mathcal{Q}_\beta^{p^*, q^*}} = \Phi^{-1}(\Psi^{-1}(\bar{X}_{l-1}^{\mathbb{S}_{|\beta|}^{q^*} \times \mathbb{L}_\beta^{p^*}}))$

9: **end for**

10: **return** the final representation $X_L^{\mathcal{Q}_\beta^{p^*, q^*}}$

---

---

**Algorithm 3** $L$-layer $\mathcal{Q}$-GT learning process

---

**Input**: An input graph $G = (V_G, E_G)$ with the initial features $X^{\mathbb{E}}$
**Output**: A hidden representation $Z$

1: Determining the embedding manifold $\mathcal{Q}_\beta^{p^*, q^*}$ by using Algorithm 1
2: Projecting initial features $X_0^{\mathcal{Q}_\beta^{p^*, q^*}} = \exp_{\mathbf{o}}^\beta([0 \| f(X^{\mathbb{E}})])$
3: $X_0^{\mathcal{Q}_\beta^{p^*, q^*}} \leftarrow \sigma_{\text{LayerNorm}}(X_0^{\mathcal{Q}_\beta^{p^*, q^*}})$
4: $X_0^{\mathcal{Q}_\beta^{p^*, q^*}} \leftarrow \text{FeatureTransform}(X_0^{\mathcal{Q}_\beta^{p^*, q^*}})$
5: $X_0^{\mathcal{Q}_\beta^{p^*, q^*}} \leftarrow \sigma_{\text{Activation}}(X_0^{\mathcal{Q}_\beta^{p^*, q^*}})$
6: $X_0^{\mathcal{Q}_\beta^{p^*, q^*}} \leftarrow \sigma_{\text{Dropout}}(X_0^{\mathcal{Q}_\beta^{p^*, q^*}})$
7: **for** $l = 1$ to $L$ **do**
8: $\quad \hat{X}_{l-1}^{\mathcal{Q}_\beta^{p^*, q^*}} = \sigma_{\text{LayerNorm}}(X_{l-1}^{\mathcal{Q}_\beta^{p^*, q^*}})$
9: $\quad \tilde{X}_{l-1}^{\mathcal{Q}_\beta^{p^*, q^*}} = \text{LinearAttention}(\hat{X}_{l-1}^{\mathcal{Q}_\beta^{p^*, q^*}})$
10: $\quad \bar{X}_{l-1}^{\mathcal{Q}_\beta^{p^*, q^*+1}} = \tilde{X}_{l-1}^{\mathcal{Q}_\beta^{p^*, q^*}} \oplus_\beta X_{l-1}^{\mathcal{Q}_\beta^{p^*, q^*}}$
11: $\quad \bar{X}_{l-1}^{\mathcal{Q}_\beta^{p^*, q^*}} = \text{FeatureTransform}(\bar{X}_{l-1}^{\mathcal{Q}_\beta^{p^*, q^*+1}})$
12: $\quad \breve{X}_{l-1}^{\mathcal{Q}_\beta^{p^*, q^*}} = \sigma_{\text{Activation}}(\bar{X}_{l-1}^{\mathcal{Q}_\beta^{p^*, q^*}})$
13: $\quad X_l^{\mathcal{Q}_\beta^{p^*, q^*}} = \sigma_{\text{Dropout}}(\breve{X}_{l-1}^{\mathcal{Q}_\beta^{p^*, q^*}})$
14: **end for**
15: $X_L^{\mathcal{Q}_\beta^{p^*, q^*}} \leftarrow \text{FeatureTransform}(X_L^{\mathcal{Q}_\beta^{p^*, q^*}})$
16: $X_{L'}^{\mathcal{Q}_\beta^{p^*, q^*}} = L'\text{-layer } \mathcal{Q}\text{-GCN2}(X_0^{\mathcal{Q}_\beta^{p^*, q^*}})$
17: $\mathbf{X}_1 = \log_{\mathbf{o}}^\beta(\Phi(X_L^{\mathcal{Q}_\beta^{p^*, q^*}}))$
18: $\mathbf{X}_2 = \log_{\mathbf{o}}^\beta(\Phi(X_{L'}^{\mathcal{Q}_\beta^{p^*, q^*}}))$
19: $\bar{\mathbf{X}} = \alpha\mathbf{X}_1 + (1 - \alpha)\mathbf{X}_2$
20: $Z = \exp_{\mathbf{o}}^\beta(\bar{\mathbf{X}})$
21: **return** the final representation $Z$

---

