# OpenReview forum: "Pseudo-Riemannian Graph Transformer"
_NeurIPS.cc/2025/Conference — NeurIPS 2025 poster_

### Official Review · Reviewer_AG9m · 2025-06-04

**Clarity:** 2
**Significance:** 3
**Originality:** 1
**Rating:** 4
**Confidence:** 4

**Summary:**

The paper presents a novel graph transformer that is grounded on a pseudo-Riemannian manifold as the underlying space. By exploiting the properties of network spherical embedding and network hyperbolic embedding, the proposed approach constructs a geometric embedding model capable of preserving both cyclical and hierarchical graph representations. Experimental results show that this method achieves superior performance compared to the traditional Graph Convolutional Network (GCN) in node classification and link prediction tasks.

**Questions:**

1. The authors could clarify the meaning of "x_0" in line 99 and Equation (2), as this is crucial for understanding the definition of the "Zero-first submanifold," which plays a central role in the paper.

2. The authors should explain how the two mappings introduced in the paper address the "geodesic disconnections issue," as this would provide more insight into this important aspect.

3. The specific meaning of the notation "x" and "z" in Equation (4) should be clarified, as this would help readers better understand the formalism and its application.

4. The conclusion of Theorem 1 seems somewhat self-evident, given that a pseudo-Riemannian manifold can be decomposed into the Cartesian product of subspaces formed by spherical and hyperbolic spaces. The focus should perhaps be placed on demonstrating the geodesics for these two spaces, specifically proving that a geodesic exists in the pseudo-Riemannian manifold composed of paths from the two subspaces constructed by the authors. This would strengthen the contribution of the theorem.

**Ethical Concerns:**

["NO or VERY MINOR ethics concerns only"]

**Final Justification:**

The authors have satisfactorily resolved my doubts and presented a persuasive argument, which has allowed me to appreciate the innovative aspects of their manuscript.

**Limitations:**

The authors have discussed the limitations of their method in practical applications. However, they have not illustrated some inherent theoretical limitations of the approach, such as the completeness of the embedding in the pseudo-metric space and the lack of interpretability. These aspects could be important for a more comprehensive evaluation of the method.

**Paper Formatting Concerns:**

1. In line 119, there seems to be an issue with the punctuation before "followed."
2. The references, e.g., (Gu, A.; Sala, F.;), (Bachmann, G.; Bécigneul,), and et al., do not include page numbers. The authors are kindly requested to carefully check and ensure that all necessary page numbers are included for these references.
3. There seems to be an issue with the formatting of Equations (9)–(12).

**Quality:**

2

**Strengths And Weaknesses:**

Strengths:

The work covers a broad range of topics, from the foundational theory of graph embedding to downstream applications. The research process is comprehensive, and the objective is to address the issue of selecting the underlying embedding space for graph embeddings, which holds significant research value.

Weaknesses:

1. Theoretical Limitations:

The authors have not demonstrated the theoretical validity of the claims made in the paper regarding the network pseudo-Riemannian manifold embedding. Specifically:

    (a) There is no theoretical proof that the method described in "Contributions 1" preserves the cyclical and hierarchical representations of graphs as claimed.

    (b) The issue of "geodesic disconnections" that the authors frequently emphasize remains unresolved.

2. Experimental Limitations:

    (a) The authors' choice of comparison methods is limited, mainly focusing on GCN and its hyperbolic variant, HGCN.

    (b) The ability of the proposed method to "capture mixed-topological dependencies" has not been empirically validated.

---

> ### Author Rebuttal · Authors · 2025-07-30
>
> We would like to thank Reviewer AG9m for the constructive comments, we would like to address the raised weaknesses and questions as below:
> ## Weakness 1:
>
> (a) Establishing a theoretical proof for model properties in non-Euclidean spaces remains  an open research problem, especially in the context of pseudo-Riemannian geometry.
> Nevertheless, among existing pseudo-Riemannian approaches, our method stands out as the most effective in both interpreting the behavior of pseudo-hyperboloids and identifying optimal configurations for the input data.
> In our Experiments (Section 6 in the main paper), the promising performance of $Q$-GT and $Q$-GCN2 provides evidence of their effectiveness in capturing and preserving cyclical and hierarchical patterns within an input graph.
> For example, on spherical datasets such as Cora and Arxiv, both models outperform existing baselines across downstream tasks.
> Additionally, the visualizations in Figure 3 (Section 6) and Figure 5 (Appendix H.3) highlight their capability to differentiate between high-order and low-order nodes, effectively capturing the underlying hierarchical organization.
>
> (b)
> In pseudo-hyperboloid $Q_{\beta}^{p,q}$, geodesic disconnection occurs when no smooth geodesic exists between two points $x, y$, resulting in an undefined logarithmic map [1], [2].
> As a result, pseudo-Riemannian methods are unable to project graph data onto the tangent space to perform graph operations, as is commonly done in hyperbolic models.
> Our approach conducts an extrinsic mapping $\Phi$ and a diffeomorphism $\Psi$ to transform the pseudo-hyperboloid $Q_{\beta}^{p,q}$ into the product of spherical and hyperbolic manifolds, each of which is geodesic connected—meaning any pair of points is joined by a smooth geodesic path.
> Our method operates directly on the product manifold, enabling better preservation of embedding quality while reducing computational complexity.
>
> ## Weakness 2:
>
> (a) We respectfully disagree with the opinion that our work mainly focus on GNNs.
> In fact, our study gives balanced attention to both GNNs and Transformer-based models.
> This is evident from the baseline selection, excluding $Q$-GCN2 (which is also our proposed method), the number of GNN and Transformer baselines is equal.
> Furthermore, SGFormer and HypFormer are among the strongest baselines in Euclidean and hyperbolic spaces, respectively.
> Our ablation study focuses on $Q$-GCN because it represents the most recent and directly relevant pseudo-Riemannian GNN approach.
> During the rebuttal phase, we extended our comparisons by including the Transformer-based method FPS-T and conducting additional ablation studies across both GNN and Transformer models in Tables 1, 2, and 3 (in response to Reviewer 8CPi).
> These results further validate the effectiveness of our proposed approach.
>
> (b)
> As discussed in Weakness 1a, both $Q$-GT and $Q$-GCN2 demonstrate strong performance on real-world datasets characterized by mixed topological structures across multiple downstream tasks.
> To assess their ability to capture hierarchical patterns, we introduce tree-structured datasets in Appendix H.3.
> The resulting visualizations reveal distinct clustering of low-level nodes, indicating that both models effectively preserve the hierarchical nature of the data.
> However, conducting more in-depth experiments to illustrate how the models capture spherical and hyperbolic patterns is currently limited by the lack of advanced visualization tools for pseudo-Riemannian spaces, which remains as an open research challenge.
> As part of future work, we plan to develop a dedicated pseudo-Riemannian visualization tool to further explore and explain the geometric behavior of $Q$-GT and $Q$-GCN2.
>
> ## Question 1:
>
> The term $x_0$ refers to the first coordinate of a point $x = (x_0, x_1, ..., x_{n-1}) \in \mathbb{R}^n$.
> In line 99, $x_0$ appears within the context of the Lorentz model for hyperbolic space, where the first coordinate is required to be positive.
> In contrast, in Equation (2), $x_0$ is part of the definition of the zero-first submanifold $\hat{Q}_{\beta}^{p,q+1}$, which imposes the condition that the first coordinate of any point $x$ on the zero-first submanifold must be zero.
> The definition of the "zero-first submanifold" is central to our work, as it serves as an intermediary space that enables the decomposition of the pseudo-hyperboloid into a product of spherical and hyperbolic manifolds.
> This submanifold also facilitates the definition of our pseudo-hyperboloid vector addition.
>
> ## Question 2:
>
> We explain how our diffeomorphic framework address the geodesic disconnection issue in our above response in Weakness 1b.
> The extrinsic function $\Phi$ and the diffeomorphism $\Psi$ transform a pseudo-hyperboloid $Q_{\beta}^{p,q}$ into the product of spherical and hyperbolic manifolds.
> Both spherical and hyperbolic manifolds are geodesic connected in our work, which enables either tangent space projection or direct learning on the manifolds.
> As stated in the Introduction (Section 1), our diffemorphism effectively addresses the issue of geodesic disconnection and preserves the cyclical and hierarchical structures of graphs.
>
> ## Question 3:
>
> We would like to thank you for your feedback.
>
> In Equation (4), the variable $x$ denotes a point on the zero-first submanifold  $\hat{Q}_{\beta}^{p,q+1}$ , where the first coordinate $x_0$ is constrained to be zero.
>
> The next $q+1$ coordinates are represented by $u = (x_1, ..., x_{q+1})$, and the remaining $p$ coordinates by $v = (x_{q+2}, ..., x_{q + p + 1})$.
>
> Meanwhile, the notation $z = (u', v')^T$ refers to a point on the product manifold , composed of two components $u' \in \mathbb{S}^q_{|\beta|}$, $v' \in \mathbb{L}^{p}_{\beta}$.
>
> The point $z$ can be interpreted as the corresponding representation of $x$ on the product manifold.
>
> ## Question 4:
>
> We do not agree that Theorem 1 is straightforward and we would like to emphasize that this work is the first to decompose pseudo-Riemannian manifold into the product of spherical and hyperbolic spaces.
> Importantly, the transformation from a pseudo-hyperboloid $Q_{\beta}^{p,q}$ to the product space of spherical and hyperbolic manifolds cannot be done directly; it requires an extrinsic mapping to ensure the existence of a valid diffeomorphism.
> According to differential geometry theory, a diffeomorphism between two manifolds $A$ and $B$ exists only if both share the same dimensionality [3].
> For any pair of points on a pseudo-hyperboloid—regardless of whether they are geodesic disconnected—the diffeomorphic transformation ensures that they become connected by both a spherical and a hyperbolic paths due to the connection property of the product space.
> However, a geodesic in a pseudo-hyperboloid corresponds to only one of three geometric types (hyperbolic, flat, or spherical) [1], [2].
> Therefore, it is counterintuitive to assume that a single geodesic in a pseudo-hyperboloid composed of both hyperbolic and spherical paths in the product space.
> This is precisely why we do not use the distance metric defined in the product space.
> Instead, we rely on the intrinsic geodesic distance of the pseudo-hyperboloid, particularly in link prediction task.
> We appreciate your suggestions for further clarification and expansion of the theoretical contributions in our work.
> We propose an alternative and promising direction for theoretical analysis of our diffeomorphic framework is to examine how specific geometric relationships are preserved in the product space.
> For instance, given a pair of nodes connected by a hyperbolic geodesic in the pseudo-hyperboloid, one could investigate how this relationship is represented after mapping to the product space, and whether the hyperbolic component contributes more significantly to model performance than the spherical one.
>
> ## Paper Formatting Concerns:
>
> Thank you for your suggestion on the paper formatting.
> We will change the punctuation mark in line 119 and update the page numbers for references in the next version.
> We would also revise the formatting issues in Equations (9)–(12) in the paper as suggestion.
>
> [1] Law, M. T.; Stam, J. "Ultrahyperbolic Representation Learning". Neural Information Processing Systems. (2020).
>
> [2] Law, M. T. "Ultrahyperbolic Neural Networks". Neural Information Processing Systems. (2021).
>
> [3] Joel W. Robbin, Dietmar A. Salamon. "Introduction to Differential Geometry" book, p413 (2024).

---

> > ### Comment · Reviewer_AG9m · 2025-08-01
> > **Regarding the geodesic problem of the constructed product space**
> >
> > (1) Regarding Question 4, I believe the dimensional condition for the existence of a diffeomorphism is relatively easy to satisfy. Specifically, it suffices that the sum of the dimensions of the individual subspaces equals that of the original space. Moreover, since there is no explicit constraint imposed on the dimensionality of the embedding space for the graph, the construction of a diffeomorphic mapping can be naturally achieved without concern for the dimensionality of the target space.
> >
> > (2) Although the authors provide substantial discussion on geodesics under diffeomorphisms, I still have some concerns. The manuscript claims that “a geodesic in a pseudo-hyperboloid corresponds to only one of three geometric types,” and it further establishes a diffeomorphic mapping between the pseudo-hyperboloid and the product space. Given that a diffeomorphism preserves connectedness and path-connectedness, one may consider the following scenario: suppose one mapped point lies in the spherical subspace and the other in the hyperbolic subspace. How, then, is the geodesic between these two points defined in the product space? In particular, how is the type of geodesic determined under such a mixed subspace setting?

---

> > > ### Author Response · Authors · 2025-08-02
> > > **Comments about geodesic problem**
> > >
> > > Dear Reviewer AG9m,
> > >
> > > We thank you for your constructive feedback on the discussion.
> > > We would like to address the concerns you raised as follows:
> > >
> > > (1) While we acknowledge that satisfying the dimensionality requirement for a diffeormophism relatively straightforward since the product space $\mathbb{S}^q_{|\beta|} \times \mathbb{L}^p_{\beta}$ has one more dimension than the pseudo-hyperboloid $Q_{\beta}^{p,q}$, our key contribution lies in extending the pseudo-hyperboloid by one dimension and constructing a diffeomorphism accordingly.
> > > We propose an extrinsic mapping $\Phi$ that projects to a zero-first submanifold, by appending a zero coordinate to the prefix of node embeddings, thus ensuring compliance with the pseudo-hyperboloid's definition.
> > > This submanifold not only retains the initial features but also facilitates the formulation of vector addition in pseudo-Riemannian space, which is an essential operation for our proposed $Q$-GT.
> > >
> > > (2)
> > > According to the definition of a product space $\mathbb{S} \times \mathbb{L}$, any point $x = (u, v)$ in this space is represented by two components: $u \in \mathbb{S}$ and $v \in \mathbb{L}$ - with the coordinates of $x$ split across the two manifolds $\mathbb{S}$, $\mathbb{L}$ [1].
> > > When a point from a pseudo-hyperboloid is mapped to the product space, it is simultaneously represented on both a spherical and a hyperbolic manifold.
> > > In this setup, the geodesic between two points $x = (u_1, v_1)$ and $y = (u_2, v_2)$ in the product space $\mathbb{S} \times \mathbb{L}$ is described by two separate paths: $\gamma_S(u_1, u_2)$ in the spherical space and $\gamma_L(v_1, v_2)$ in the hyperbolic space.
> > > Consequently, a geodesic $\gamma$ on the pseudo-hyperboloid becomes a combination of $\gamma_S$ and $\gamma_L$.
> > >
> > > Distances in the product space $\mathbb{S} \times \mathbb{L}$ are typically defined using either the $l_1$ or $l_2$ norms that sum the lengths of the component paths, or the $\min$ distance which selects the shorter one.
> > > However, as noted in [2], enforcing both spherical and hyperbolic structures equally via $l_1$ or $l_2$ distances may contradict the nature of the pseudo-hyperboloid.
> > > Similarly, the $\min$ distance can be biased toward one component while neglecting the other.
> > > In contrast, the intrinsic distance on the pseudo-hyperboloid dynamically adapts to the inner product between points, naturally selecting the appropriate geometric behavior (spherical or hyperbolic).
> > > For this reason, in our framework, the product space is used primarily for performing graph transformations such as linear attention and layer normalization, not for computing distances.
> > >
> > > [1] Albert Gu, Frederic Sala, Beliz Gunel, Christopher Ré. "Learning Mixed-Curvature Representations in Product Spaces". International Conference on Learning Representations (2019).
> > >
> > > [2] Law, M. T. "Ultrahyperbolic Neural Networks". Neural Information Processing Systems. (2021).

---

> > > > ### Comment · Reviewer_AG9m · 2025-08-02
> > > >
> > > > Thank you for your clarification. My concerns have been resolved, and I will update my rating to 4.

---

> > > > > ### Author Response · Authors · 2025-08-02
> > > > >
> > > > > Thank you once again for taking the time to review our paper and participate in the rebuttal discussion. We’re glad to hear that you found our response helpful. If you have any additional feedback or questions, we would be happy to address them and revise the draft accordingly. We truly appreciate your time and effort!

---

### Official Review · Reviewer_V9gp · 2025-07-02

**Clarity:** 3
**Significance:** 3
**Originality:** 3
**Rating:** 5
**Confidence:** 3

**Summary:**

This paper addresses the challenge of representing complex graph structures that are poorly captured by manifolds of uniform global curvature (e.g., hyperbolic or spherical spaces). The authors introduce a diffeomorphic framework for graph embedding on pseudo-Riemannian manifolds, generalizing both hyperbolic and spherical geometries. Building on this, they propose the pseudo-Riemannian Graph Transformer (Q-GT), which extends transformer operations (attention, residual connection, layer norm) to pseudo-Riemannian geometry. They also design a lightweight space searching algorithm to select the best manifold for a given graph. Experiments on node classification and link prediction across diverse real-world graphs demonstrate that Q-GT outperforms state-of-the-art baselines.

**Questions:**

1) How does the Q-GT compare in terms of training/inference time, parameter count, and GPU memory usage to the strongest baselines (e.g., Hypformer, SGFormer, Q-GCN)?
2) Could the authors conduct the experiments on larger graphs such as Open Graph Benchmark?
3) Which components of the Q-GT are most critical for its success (e.g., diffeomorphic mapping, specific attention mechanism, residual connection)? Is there any interpretability in the learned embeddings (e.g., correlation between curvature and graph properties)?

**Ethical Concerns:**

["NO or VERY MINOR ethics concerns only"]

**Final Justification:**

My concerns have been addressed and I increased my score.

**Limitations:**

Yes: The authors have adequately discussed limitations

**Quality:**

3

**Strengths And Weaknesses:**

Strengths:
1) The Q-GT is the first transformer-based graph model built in pseudo-Riemannian geometry, enabling core transformer operations in this space.
2) The space searching algorithm is simple yet but grounding the manifold selection in data-driven curvature estimation.
3) Q-GT achieves strong, consistent performance on both node classification and link prediction tasks, outperforming established baselines such as QGCN (see Tables 1 and Table 2).

Weakness:
1) The space searching algorithm is grounded in a linear relationship between ideal standard deviation and space dimension, but this is mostly heuristic and lacks theoretical justification or sensitivity analysis.
2) The paper lacks some analysis on training/inference time and scalability or complexity. Q-GT introduces higher computational and memory overhead compared to GNNs (and possibly some baselines), which might be a practical concern.
3) The method uses space search, which takes more computation time and it is not fair to compare with other approaches without space search such as QGCN.

---

> ### Author Rebuttal · Authors · 2025-07-30
>
> We would like to thank Reviewer V9gp for the constructive comments, we would like to address the raised weaknesses and questions as below:
>
> ## Weakness 1:
> We agree with Reviewer V9gp that our work requires the assumption of a linear relationship between the ideal standard deviation $\sigma_i$ and the space dimension $p_i$.
> In general, the ideal standard deviation can be modeled as any bijective function of $p_i$, such as a quadratic function $p_i^2$ or an exponential form $e^{p_i}$, as long as it aligns with the curvature behavior of the three geometric settings (i.e., monotonically increasing in spherical space and inversely related in hyperbolic space; see Appendix C.3 for details).
> However, using alternative forms like the quadratic $p_i^2$ results in a steeper rate of change, which may lead to suboptimal manifold selection.
> To clarify this hypothesis, we define the ideal standard deviation $\sigma_i$ using a quadratic formulation as follows:
> $$
>     \sigma_i = \begin{cases}
>       \sigma_{\text{max}} - (\sigma_{\text{max}} - \sigma_{\text{min}}) \left(\frac{2p_{i} - (d + 1)}{d-3} \right)^2    & \text{if $ (d-1)/2 < p_i \leq d-1$} ; \\
>      1  & \text{if $p_i = (d-1)/2$} ; \\
>      \sigma_{\text{min}} + (\sigma_{\text{max}} - \sigma_{\text{min}}) \left(\frac{2p_{i}}{d-3} \right)^2 & \text{if $ 0  \leq p_i < (d-1)/2$}.
>     \end{cases}
> $$
> Then we apply the space searching algorithm in Section 5 to four benchmark datasets: Cora, Citeseer, Airport, and Pubmed.
> The resulting space dimensions are $p_i = 4, 7, 2, 4$, respectively.
> However, when compared to the configurations shown in Fig. 2 (Section 6) and Fig. 4 (Appendix H), these values do not correspond to the optimal configurations for pseudo-hyperboloids.
> This suggests that the assumption of a linear rate of change in standard deviation aligns well with the variation in space dimension $p_i$.
> While further theoretical justification is needed, our experimental results provide strong evidence that the linear assumption is effective in practice.
> A more detailed theoretical analysis would require a proper model of the underlying graph manifold in real world dataset, which can be considered as a future work.
>
> ## Weakness 2:
>
> **Table 4.** Comparison of training time (in seconds).
>
> | Dataset         | SGFormer | Hypformer | $Q$-GCN | $Q$-GCN2 | $Q$-GT  |
> |-----------------|----------|-----------|--------|---------|--------|
> | Cora            | 40.92    | 66.42     | 136.6  | 82.44   | 193.7  |
> | Citeseer        | 40.21    | 67.31     | 141.8  | 76.12   | 188.2  |
> | Airport         | 71.55    | 97.20     | 211.4  | 66.59   | 249.5  |
> | Pubmed          | 144.8    | 153.8     | 278.0  | 83.52   | 386.0  |
> | Arxiv           | 100.1    | 136.7     | 347.2  | 145.9   | 353.5  |
> | Penn94          | 65.49    | 140.1     | 299.7  | 121.4   | 258.5  |
> | Twitch Gamers   | 163.3    | 148.7     | 377.2  | 181.2   | 386.3  |
> | Products        | 354.1    | 371.8     | 461.5  | 325.2   | 681.2  |
> | Vessel          | 994.5    | 931.4     | 1137.2 | 664.0   | 1024.8 |
> | **Avg Rank**    | **1.6**  | **2.1**   | **3.8**| **1.7** | **4.2**|
>
> **Table 5.** Comparison of GPU memory usage (in MiB).
>
> | Dataset         | SGFormer | Hypformer  | $Q$-GCN | $Q$-GCN2 | $Q$-GT  |
> |-----------------|----------|-----------|--------|--------|--------|
> | Cora            | 58.4     | 158.6     | 111.7  | 65.9   | 70.8   |
> | Citeseer        | 126.8    | 471.2     | 314.8  | 134.5  | 129.6  |
> | Airport         | 42.8     | 57.9      | 107.3  | 58.4   | 82.0   |
> | Pubmed          | 195.5    | 382.9     | 428.2  | 240.2  | 348.6  |
> | Arxiv           | 1290.1   | 3822.7    | 4536.0 | 1106.2 | 3098.1 |
> | Penn94          | 2246.6   | 7281.5    | 4876.2 | 2319.9 | 1585.1 |
> | Twitch Gamers   | 1638.4   | 1578.0    | 2100.2 | 2840.0 | 2713.5 |
> | Products        | 27979.5  | 32363.3   | 58954.8| 34668.1| 51574.0|
> | Vessel          | 8016.3   | 17376.8   | 48672.1| 14249.9| 39293.8|
> | **Avg Rank**    | **1.1**  | **3.4**   | **4.4**| **2.7**| **2.7**|
>
> In our paper, we provided a runtime comparison of three pseudo-Riemannian models in Appendix H.2.
> To offer a more comprehensive complexity analysis, we further evaluate the runtime and memory usage of $Q$-GT against strong baselines, including SGFormer, HypFormer, and $Q$-GCN, as suggested in Question 1.
> The detailed results are presented in Table 4 and Table 5.
> A comparison between $Q$-GCN and $Q$-GCN2 shows that our diffeomorphic framework achieves better efficiency in both runtime and memory.
> This is because $Q$-GCN framework relies on a combination of diffeomorphic and tangent space mappings in product manifolds, whereas our approach operates directly within the product manifold space.
> Nonetheless, $Q$-GT incurs higher computational and memory overhead compared to GNNs, SGFormer, and HypFormer, particularly on large-scale datasets like Products and Vessel.
> We attribute this overhead to Transformer components such as linear attention, the refining function, and residual connections, which we recognize as a limitation of our method.
> However, this trade-off enables $Q$-GT to deliver strong performance on large datasets, as shown in Table 3.
> $Q$-GT delivers the best performance across both datasets, surpassing the strongest baselines by 7.3\% on Products and 32.09\% on Vessels.
> We are optimistic that future work can further reduce the model’s complexity and retain its effectiveness.
>
> ## Weakness 3:
> We would like to clarify that our space searching algorithm is introduced to identify optimal pseudo-Riemannian manifolds for graph embedding.
> While it may incur higher computational costs on large graphs, it is significantly more efficient than the brute-force approach used in prior works such as $Q$-GCN (see Section H.2 for runtime comparison).
> In node classification and link prediction tasks, we compare the results of all pseudo-Riemannian methods ($Q$-GCN, $Q$-GCN2, and $Q$-GT) using the manifold configurations computed by our algorithm.
> To ensure fairness, we also perform a brute-force search over all manifold configurations on four benchmark datasets—Airport, Cora, Citeseer, and Pubmed (see Figure 2 in Section 6 and Appendix H.1).
> The results reveal that $Q$-GCN's chosen manifold does not align with the outcome of our space search algorithm or the intrinsic topology of the input graph, and it fails to outperform $Q$-GCN2 and $Q$-GT on the optimal pseudo-hyperboloid.
>
> ## Question 1:
>
> We provided a complexity comparison in Table 4 and Table 5, as discussed in Weakness 2.
> Overall, $Q$-GT introduces higher computational and memory costs compared to other baselines.
> We acknowledge this as a shortcoming and intend to explore solutions in future work.
>
> ## Question 2:
>
> We understand that scalability is a key concern for Transformer-based models.
> To evaluate this, we tested our $Q$-GT and other Transformers on larger datasets from the Open Graph Benchmark, with the results reported in Table 3 above (in response to Reviewer 8CPi).
> Additionally, runtime and memory comparisons with other baselines are provided in our reply in Weakness 2.
> Although $Q$-GT incurs high computational complexity, this trade-off results in notably strong performance of $Q$-GT.
> We recognize that improving the complexity of $Q$-GT remains an important direction for future work.
>
> ## Question 3:
>
> The most critical component of QGT is the diffeomorphic mapping, which transforms the pseudo-hyperboloid into a product space of spherical and hyperbolic geometries.
> The diffeomorphic framework enables a more interpretable evolution of the pseudo-hyperboloid $Q_{\beta}^{p,q}$, leading to our efficient space searching algorithm for determining the optimal pseudo-hyperboloid.
> By addressing the issue of geodesic disconnection, it facilitates the adaptation of standard architectures (e.g., GNNs and Transformers) to pseudo-Riemannian spaces.
> Moreover, by preserving both spherical and hyperbolic characteristics, the diffeomorphic mapping contributes to improved performance in graph learning tasks.
> Additionally, other components such as linear attention, the refining function, and residual connections also play a crucial role in the success of $Q$-GT.
>
> To interpret the learned embeddings, we employ the UMAP tool for visualization.
> For instance, in Appendix H.3, we visualize the embeddings of tree-structured datasets on the hyperbolic component of the product space.
> The visualizations reveal that in $Q$-GCN2 and $Q$-GT, red and pink nodes—corresponding to lower-level nodes—form distinct clusters, indicating that hierarchical information is effectively preserved.
> In contrast, $Q$-GCN exhibits oversmoothing, where embeddings of low-level and high-level nodes become indistinguishable, suggesting a loss of hierarchical structure.

---

### Official Review · Reviewer_SXvr · 2025-07-03

**Clarity:** 4
**Significance:** 4
**Originality:** 4
**Rating:** 6
**Confidence:** 4

**Summary:**

Real-world graphs often exhibit complex topological structures, and hence it is often more suitable to embed graphs into pseudo-Riemannian manifolds than commonly used Euclidean space. Existing methods exhibit information loss, a lack of guideline for selecting dimensions, and limited context size, which limit their capacity. This paper proposes the pseudo-Riemannian graph transformer by extending linear attention and layer normalization to the pseudo hyperboloid. A space searching algorithm is designed based on the Gaussian Sectional Curvature distribution. Experiments corroborate the effectiveness of the proposed method.

**Questions:**

See above.

**Ethical Concerns:**

["NO or VERY MINOR ethics concerns only"]

**Final Justification:**

My concern has been addressed and I think this submission is a qualified NeurIPS paper.

**Limitations:**

Yes

**Quality:**

4

**Strengths And Weaknesses:**

Strengths:

The proposed method is based on solid theory.

The first paragraph of section 5 clearly explains what kind of graphs are suitable for the proposed pseudo-Riemannian graph transformers.

The space searching algorithm increases the practicality of the proposed model.

The visualization of the learned embeddings is informative and interesting.

Weaknesses:

In line 47, the concepts of time and space dimensions appear without definition. And where do the names come from?

---

> ### Author Rebuttal · Authors · 2025-07-30
>
> Thank you for your encouraging feedback. We would like to address the weakness points as below:
> ## Weakness 1:
> We’re pleased to hear that you value the elegance of our contributions to graph learning in pseudo-Riemannian geometry.
> In line 47, we mentioned the concepts of space and time dimensions of pseudo-Riemannian manifolds.
> A more detailed explanation is provided in Section 2 (lines 97–98).
> Given a point $x=(x_0, x_1, ..., x_{q}, x_{q+1}, ..., x_{q+p})$ on the pseudo-hyperboloid $Q_{\beta}^{p,q}$  , the first $q + 1$ coordinates $(x_0, x_1, ..., x_q)$ correspond to time dimensions, while the remaining $p$ coordinates $(x_{q+1},..., x_{q+p})$ represent space dimensions.
> The concepts of time and space dimensions originate from Einstein’s general theory of relativity which described the four-dimensional spacetime.
> The time and space dimensions are formalized within the framework of pseudo-Riemannian geometry[1], then widely adopted in pseudo-Riemannian learning methods [2], [3], [4].
> However, finding the appropriate time and space dimensions that align with the graph structure is a non-trivial and time-consuming task as it requires extensive training if done manually.
> To address this challenge, we introduces an automated space searching algorithm that efficiently discovers optimal pseudo-Riemannian configurations based on the Gaussian Sectional Curvature.
> This significantly reduces manual effort and improves model performance.
>
> [1] O’neill, B. V. "Semi-Riemannian Geometry With Applications to Relativity", 54 (1983).
>
> [2] Law, M. T.; Stam, J. "Ultrahyperbolic Representation Learning".  Neural Information Processing Systems. (2020).
>
> [3] Law, M. T. "Ultrahyperbolic Neural Networks". Neural Information Processing Systems. (2021).
>
> [4] Xiong, B.; Zhu, S.; Potyka, N.; Pan, S.; Zhou, C.; Staab, S. "Pseudo-Riemannian Graph Convolutional Networks". Neural Information Processing Systems. (2021).

---

### Official Review · Reviewer_8CPi · 2025-07-03

**Clarity:** 3
**Significance:** 2
**Originality:** 2
**Rating:** 4
**Confidence:** 4

**Summary:**

This work presents a graph transformer that operates in a Pseudo-Riemannian manifold, addressing limitations of prior hyperbolic and spherical approaches. The authors design a diffeomorphic transformer that splits the pseudo-Riemannian space into coupled hyperbolic and spherical components, thereby eliminating geometric disconnection. Building on this, they create the Pseudo-Riemannian Graph Transformer (Q-GT) and a curvature-aware search procedure that selects manifold dimensions by aligning the input graph’s Gaussian sectional-curvature (GSC) profile. Tested on seven node-classification and four link-prediction datasets, Q-GT achieves small yet consistent gains over strong Euclidean and hyperbolic baselines.

**Questions:**

- Is extending LResNet to the product space non-trivial?
- How is the UMAP used for the visualization in Poincare space? Do you first log map the embeddings then use the UMAP or do you use a hyperbolic UMAP?

**Ethical Concerns:**

["NO or VERY MINOR ethics concerns only"]

**Final Justification:**

The authors provided new experimental results, particularly on the OGB datasets, which addressed my earlier concerns regarding the scalability of the method. These additions, along with clarifications on design choices and comparisons with prior work justify the score increase.

**Limitations:**

The authors have addressed the limitations.

**Quality:**

2

**Strengths And Weaknesses:**

### Strengths
The paper is well written. Using the product manifolds retain both the cyclical and hierarchical information from the data. The proposed space search algorithm is also efficient and removes the need for manual fine-tuning significantly. The theoretical discussion is rigorous and supported by clearly presented proofs, enhancing the paper’s credibility. Experiments span homophilous and non-homophilous graphs, where Q-GT outperforms baseline in 5/7 classification tasks and all link-prediction tasks.

### Weaknesses
- Use of log map and exp map in residual connection is inefficient. Recent methods like LResNet [1] have developed a fully hyperbolic residual connection method that does not rely on mapping. This approach can effectively lower computational overhead, reduce numerical errors from repeated projections, and preserve intrinsic curvature throughout training, thereby enabling faster and more stable convergence
- The paper omits comparison with the recent FPS-T [2] method, which overlaps significantly with Q-GT despite being cited. The authors should clearly address what are the differences between Q-GT and FPS-T.
- Gains over the strongest baselines (e.g., SGFormer, Hypformer) are often ≤1–2 F1 on `Cora`, `Pubmed` and `Penn94`, calling into question whether the extra geometric machinery justifies its cost.
- Most benchmark datasets are small (< 10 k nodes), with the largest having 169 k nodes. Including larger benchmarks such as `ogbn-products` and `ogbl-vessel` would better demonstrate the model’s scalability.
- Although the visualization of Q-GT embeddings in Figure 3 shows that nodes are somewhat dispersed, there is still no clear hierarchy. Is this because the dataset lacks any hierarchy or does the model require extra steps to capture the hierarchy completely? How would this compare to other hyperbolic models like Hypformer?
- "Geodesically disconnection" reads awkwardly. I am assuming the it should be "Geodesic disconnection".

Overall, I think this manuscript is promising and clearly presented, but it stops short of fully demonstrating the practical value of its geometric innovations. Adding (i) direct comparisons and clarifications versus closely related methods such as FPS-T and mixed-curvature Transformers, (ii) evaluation on large-scale benchmarks (e.g., `ogbn-products` and `ogbl-vessel`), and (iii) designing a fully Pseudo-Riemannian residual connection would substantially elevate the paper’s impact.

[1] He, Neil, Menglin Yang, and Rex Ying. "Lorentzian residual neural networks." arXiv preprint arXiv:2412.14695 (2024).
[2] Cho, Sungjun, et al. "Curve your attention: Mixed-curvature transformers for graph representation learning." arXiv preprint arXiv:2309.04082 (2023).

---

> ### Author Rebuttal · Authors · 2025-07-30
>
> We would like to thank Reviewer 8CPi for the constructive comments, we would like to address the raised weaknesses and questions as below:
> ## Weakness 1:
> We sincerely thank you for suggesting the Lorentzian residual connection [1].
> In contrast to our pseudo-hyperboloid addition in Section 4, this approach performs entirely within the hyperbolic manifold, eliminating the need for tangent space.
> However, adapting the Lorentzian residual connection to pseudo-hyperboloids $Q_{\beta}^{p,q}$ requires extending it to product manifolds, which presents a non-trivial challenge.
> Building upon the definition of Lorentzian residual connection [1],  we propose the following (straightforward) extension:
>
> $$
> w_1 x \oplus_{\beta} w_2 y = \left(
> \frac{\sqrt{|\beta|}(w_1 u_1 + w_2 u_2)}{|| w_1 u_1 + w_2 u_2 ||},\
> \frac{\sqrt{|\beta|}(w_1 v_1 + w_2 v_2)}{\left| \left| w_1 v_1 + w_2 v_2 \right| \right|_{\mathbb{L}}}
> \right)
> $$
>
> where $w_1, w_2$ are the weights, $x = (u_1, v_1)^T $, and $y = (u_2, v_2)^T$ are two points in the product space of spherical and hyperbolic manifolds.
> The work in [1] proved that the denominator in hyperbolic part $| || w_1 v_1 + w_2 v_2 ||_{\mathbb{L}} |$ has a positive lower bound when the weights are positive.
> However, this property does not hold for the spherical denominator $|| w_1 u_1 + w_2 u_2 ||$, whose value can range from $0$ to $+\infty$ for any choice of $w_1$, $w_2$.
> To eliminate numerical instability in the spherical case, additional constraints may be required but potentially increasing computational complexity.
> On the other hand, our pseudo-hyperboloid vector addition maintains feature integrity, thereby improving the model’s learning capacity over prior pseudo-Riemannian approaches.
>
> ## Weakness 2:
>
> **Table 1.** Comparison to FPS-T in node classification results (F1-score).
> | Method         | Cora             | Citeseer         | Airport           | Pubmed           | Arxiv             | Penn94           | Twitch Gamers     |
> |----------------|------------------|------------------|-------------------|------------------|-------------------|------------------|-------------------|
> | FPS-T          | 60.50 ± 0.40     | 60.70 ± 0.30     | 91.41 ± 0.17      | 76.23 ± 0.31     | OOM               | OOM              | OOM               |
> |$Q$-GT          | **82.97 ± 0.17** |  **70.40 ± 0.49**     | **95.70 ± 0.08**  | **80.37 ± 0.40**   | **66.24 ± 0.05**  | **82.77 ± 0.10** | **64.45 ± 0.07**  |
>
> **Table 2.** Comparison to FPS-T in link prediction results (ROC AUC, %).
>
> | Method         | Cora             | Citeseer         | Airport           | Pubmed           |
> |----------------|------------------|------------------|-------------------|------------------|
> | FPS-T          | 95.25 ± 0.12     | 97.17 ± 0.44     | 96.10 ± 0.13      | 92.11 ± 0.04     |
> | $Q$-GT       | **96.87 ± 0.19** | **97.59 ± 0.11** | **96.56 ± 0.07**  | **96.40 ± 0.10** |
>
> Similar to $Q$-GT, FPS-T [2] is a graph Transformer designed for graphs with mixed structural properties, which becomes the concerns of the reviewers.
> FPS-T operates in a product space of Riemannian stereographic manifolds  $\lbrace st^{i} \rbrace^{H}_{i=1}$,
> where each component $st^{i}$ has a trainable curvature $\kappa_i$ that can be positive, negative, or zero.
> To address this concern, we conduct a comparative evaluation of $Q$-GT and FPS-T on both node classification and link prediction tasks, with the results presented in Table 1 and Table 2.
> For fairness, we set $H=2$ since our product space also has two components.
> First, FPS-T incurs significant computational overhead due to the use of tangent mappings and the costly computation of Laplacian eigenvectors.
> As a result, it runs out of memory on three large-scale datasets: Arxiv, Penn94, and Twitch Gamers.
> Second, the model's trainable curvatures $\kappa_i$ fail to converge to optimal values.
> On node classification tasks for Cora and Citeseer, FPS-T underperforms compared to $Q$-GT.
> Specifically, the learned curvatures $(\kappa_1, \kappa_2)$ converge to $(0.023, 0.025)$ on Cora and $(0.008, 0.007)$ on Citeseer, which do not reflect the hyperbolic geometry.
> In contrast, $Q$-GT benefits from operating directly on the product of spherical and hyperbolic manifolds, which is diffeomorphic to the pseudo-hyperboloid, thereby enhancing its performance.
>
> ## Weakness 3:
>
> To address concerns about the computational cost of $Q$-GT, we provided a detailed complexity comparison in Table 4 and Table 5 below (in response to Reviewer V9gp).
> In terms of runtime, $Q$-GT requires significantly more time than SGFormer and HypFormer.
> However, its memory usage is comparable to these models on the Cora and Pubmed datasets.
> Notably, on Penn94, $Q$-GT achieves the lowest memory consumption among all baselines.
> We also evaluated $Q$-GT against strong baselines on larger datasets (Products and Vessel from the Open Graph Benchmark) as reported in Table 3.
> While $Q$-GT is indeed more demanding in terms of both runtime and memory, it delivers substantially superior performance compared to SGFormer and HypFormer, demonstrating the value of the proposed geometric framework.
>
> ## Weakness 4:
>
> **Table 3.** Comparison on Open Graph Benchmark (OGB) datasets.
>
> | Method         | Products         | Vessel           |
> |----------------|------------------|------------------|
> | $Q$-GCN          | 29.56 ± 0.31     | 59.64 ± 0.37     |
> |$Q$-GCN2         | 52.08 ± 0.45     | 50.21 ± 0.36     |
> | SGFormer       | 65.47 ± 0.28     | 50.03 ± 0.26     |
> | Hypformer      | 40.25 ± 0.54     | 49.87 ± 0.49     |
> | $Q$-GT      | **70.25 ± 0.27** | **78.78 ± 0.50** |
>
> To further assess the scalability of our model, we conducted additional evaluations of $Q$-GT and other strong baselines on two recommended datasets from the Open Graph Benchmark: 'ogbn-products' (Products) and 'ogbl-vessel' (Vessels), where the results are shown in Table 3.
> All experiments were performed with all Transformer models configured with 1 layer, GNNs with 3 layers, and training conducted over 200 epochs.
> It is obvious that $Q$-GT achieves the highest performance on both datasets, outperforming the strongest baselines by 7.3% on Products and 32.09% on Vessel.
> However, as shown in Table 4 and Table 5, $Q$-GT introduces higher computational and memory costs compared to other baselines.
> We recognize this as a limitation of our current approach and plan to investigate more efficient solutions in future work.
>
> ## Weakness 5:
>
> As depicted in Figure 3 (Section 6 in the main paper), high-level nodes appear more scattered, forming several small clusters that include only a few low-level nodes, suggesting the presence of localized hierarchical patterns.
> However, the overall hierarchical structure is not clearly visible in this visualization, because the Cora dataset lacks a well-defined hierarchy, as reflected by its Gaussian Sectional Curvature (GSC) distribution, which tends to be more spherical.
> To better illustrate the model's ability to capture hierarchical structure, we introduce three synthetic tree-structured datasets in Appendix H.3.
> The visualizations of these datasets reveal distinct clusters of low-level nodes (represented by red and pink) in $Q$-GT and $Q$-GCN2, demonstrating the hierarchy of the datasets.
> Additionally, we compare to hyperbolic baseline, Hypformer on these tree datasets.
> Hypformer achieves average AUC scores of 95.10, 92.37, and 91.06 on Tree-1, Tree-2, and Tree-3, respectively.
> These results are comparable to $Q$-GT and $Q$-GCN2 , further confirming that both models are effective in capturing and representing hierarchical structures.
>
> ## Weakness 6:
>
> We appreciate your feedback and will update the terminology in the next version of the paper.
> The term "geodesic disconnection" is originated from the work $Q$-GCN.
>
> ## Question 1:
>
> We discussed about extending Lorentzian residual connection [1] in our reply in Weakness 1.
> We concluded that adapting Lorentzian residual connection to product space is a non-trivial challenge due to the numerical instability in spherical denominator.
>
> ## Question 2:
>
> Given the final representations of $Q$-GT and $Q$-GCN2, we first apply our diffeomorphic mapping to transform them into the product space and extract the hyperbolic components.
> We then use hyperbolic UMAP with  'output\_metric=hyperboloid' to embed these features into a 2D hyperboloid.
> In Figure 3 (Section 6 in the paper), we further project these hyperbolic outputs onto the Poincaré disk, since the original embeddings lie in a 2D hyperboloid space.
> In contrast, for the tree datasets shown in Appendix H.2, we directly visualize the hyperbolic UMAP outputs without projection to the Poincaré disk, as this yields clearer visualizations.
> We avoid applying the logarithmic map before UMAP, as tangent space mappings can introduce distortions in the quality of the hyperbolic embeddings [1].
>
> [1] He, Neil, Menglin Yang, and Rex Ying. "Lorentzian residual neural networks." arXiv preprint arXiv:2412.14695 (2024).
>
> [2] Cho, Sungjun, et al. "Curve your attention: Mixed-curvature transformers for graph representation learning." arXiv preprint arXiv:2309.04082 (2023).

---

> > ### Comment · Reviewer_8CPi · 2025-08-03
> >
> > Thank you for the detailed response. The extension of the Lorentzian residual connection to the pseudo-Riemannian space is well justified, and the additional OGB results and FPS-T comparison significantly strengthen the paper. I am satisfied with the clarifications and am increasing my score to a weak accept. Please ensure these revisions are incorporated into the final manuscript.

---

> > > ### Author Response · Authors · 2025-08-03
> > >
> > > Thank you for reviewing our paper and engaging in the rebuttal process. We’re pleased that our response was helpful, and we welcome any further comments or questions you may have. We are committed to addressing your feedback and updating the draft as needed. Your time and effort are sincerely appreciated!

---

### Decision · Program_Chairs · 2025-09-17

**Decision:**

Accept (poster)

**Comment:**

All the reviewers agree that this is a solid and meaningful contribution that advances the state-of-the-art in geometric machine learning. In some sense it's a natural evolution of continually improving and refining earlier geometric ideas and methods -- the use of hyperbolic and spherical manifolds, geometrically inspired regularization terms in representation learning, etc. -- this time by merging hyperbolic and spherical manifolds (negative and positive curvature spaces), generalizing concepts from transformers to pseudo-Riemannian spaces, and taking into account curvature to speed up learning. The generalization and synthesis of these ideas to manifolds, and validation on a wide variety of datasets, represents a meaningful contribution. Geometric methods in machine learning have now entered the mainstream conversation, and now it is no longer enough to just present a novel application of geometric methods to a machine learning problem -- real progress over the state-of-the-art is essential. This paper appears to clear that bar.